# A quantitative pipeline for whole-mount deep imaging and analysis of multi-layered organoids across scales

Alice Gros[1†§], Jules Vanaret[1,2†§], Valentin Dunsing-Eichenauer[1†], Agathe Rostan[1], Philippe Roudot[2,3], Pierre-François Lenne[1], Léo Guignard[1*‡], Sham Tlili[1*‡]

[1]Aix Marseille Univ, CNRS IBDM (UMR 7288), Turing Centre for Living Systems, Marseille, France; [2]Aix Marseille University, CNRS, Centrale Marseille, I2M (UMR 7373), Turing Centre for Living Systems, Marseille, France; [3]Aix Marseille University, CNRS, Centrale Marseille, FRESNEL (UMR 7249), Turing Centre for Living Systems, Marseille, France

**\*For correspondence:**
leo.guignard@univ-amu.fr (LG);
sham.tlili@univ-amu.fr (ST)

[†]These authors contributed equally to this work
[‡]These authors also contributed equally to this work

[§]The order of these authors reflects the amount of time each dedicated to the project

**Competing interest:** The authors declare that no competing interests exist.

## eLife Assessment

This work describes the establishment of an image analysis pipeline for signal correction, segmentation and quantitative data analysis of multilayered organoid and tumoroid systems. The revised study is **important** for the field to address many practical challenges in deep-tissue visualization. The image analysis pipeline is well-designed and **compelling**.

**Abstract** Whole-mount 3D imaging at the cellular scale is a powerful tool for exploring complex processes during morphogenesis. In organoids, it allows examining tissue architecture, cell types, and morphology simultaneously in 3D models. However, cell packing in multilayered organoid tissues hinders both deep imaging and quantification of cell-scale processes. To address these challenges, we developed an experimental and computational pipeline to extract properties at scales ranging from cell to tissue. The experimental module is based on two-photon imaging of immunostained organoids. The computational module corrects for optical artifacts, performs accurate 3D nuclei segmentation and reliably quantifies gene expression. We provide the computational module as a user-friendly Python package called Tapenade, along with napari plugins which enable joint data processing and exploration across scales. We demonstrate the pipeline by quantifying 3D spatial patterns of gene expression and nuclear morphology in gastruloids, revealing how local cell deformations and gene co-expression relate to tissue-scale organization. This quantitative pipeline improves our understanding of gastruloid development, and lays the groundwork for a wide range of multi-layered organoids and tumoroids systems

## Introduction

### Understanding organoid development and variability at multiple scales

The biology of organoids and tumoroids represents a frontier in fundamental biology and biomedical research, driven by the need for accurate 3D models complementary to 2D cell cultures and animal studies (*Clevers and Tuveson, 2019*; *Kim et al., 2020*). In the context of the '3Rs' (Replacement, Reduction, and Refinement) aimed at minimizing the use of animals in research and enhancing ethical experimental practices (*Russell et al., 1959*). Developing these models in 3D is crucial for mimicking the complex architecture and functionality of tissues, enabling deeper insights into developmental, or

disease mechanisms and treatment responses (*Simian and Bissell, 2017*; *Lancaster and Huch, 2019*). Organoids models exhibit less stereotypic development compared to embryos, where cell identification is typically straightforward based on position within the embryo at specific developmental stages (*McDole et al., 2018*; *Guignard et al., 2020*). In contrast, organoids present more complex developmental trajectories (*Farag et al., 2024*; *Suppinger et al., 2023*; *Villaronga Luque et al., 2023*; *Hofer and Lutolf, 2021*), characterized by variability in morphology, cell type composition, and differentiation levels. This variability may stem from differences in the initial cell differentiation state or number (*Cermola et al., 2021*), as well as from a less controlled biochemical and mechanical environment than that of embryos (*Gjorevski et al., 2022*), which grow confined within maternal tissues for mammals or in a rigid shell for insects (*Lenne et al., 2021*). This inherent variability underscores the need for detailed characterization of a sufficient number of organoids in parallel to properly characterize the diversity of their developmental processes. To this end, coarse-grained methods have been developed to classify phenotypes of intestinal organoids (*Lukonin et al., 2020*) and gastruloids (*Suppinger et al., 2023*; *Gritti et al., 2021*) by analyzing maximum projections from 3D immunofluorescence staining using high-throughput confocal imaging. These approaches are effective for systematically screening the impact of chemical compounds or signaling pathway perturbations on organoid morphology, as well as on the localization and proportion of various cell types and tissues within the organoid.

However, understanding how organoids or tumoroids develop and self-organize also requires capturing how individual stem cells or cancerous cells differentiate and modulate their behavior and morphology in response to their local microenvironment (*Costa et al., 2016*; *Hass et al., 2020*). Simultaneously, it is essential to quantify tissue characteristics on a coarser scale (where each measurement integrates data from groups of several neighboring cells) to capture the mechanical and biochemical interactions among different tissues and cell populations within the overall geometry of the organoid (*Liu et al., 2023*). To achieve this, a challenge is to image these 3D tissues *in toto* at a resolution sufficient to identify each cell and its neighbors both in live and fixed samples (*van Ineveld et al., 2022*). On fixed samples, multiple specific fluorescent markers can be used to capture cell fate by measuring the expression levels of particular genes or proteins by immunostaining or in situ hybridization. Complementary, live imaging is essential for understanding how local cellular events (division, migration, rearrangements) underlie the morphogenesis of these systems, as it has been done previously for embryonic development (*Bosveld et al., 2012*; *Guirao et al., 2015*).

## Imaging organoids in toto at cell resolution

Recently, light-sheet and confocal imaging systems have been optimized for parallel imaging or long-term live imaging of organoids (*Zheng et al., 2023*; *de Medeiros et al., 2022*; *Beghin et al., 2022*; *Moos et al., 2024*). Using deep-learning-based segmentation, tracking, and classification tools, these protocols generate digital atlases of developing organoids at single-cell resolution. For example, (*de Medeiros et al., 2022*) employed a dual illumination inverted light-sheet microscope to live-image gut organoids over several days, enabling cell lineage reconstruction from nuclei tracking. In (*Beghin et al., 2022*), a single-objective light-sheet microscope was used to image tens of 3D tissues, such as gut, hepatocyte, and neuroectoderm organoids, in parallel for phenotypic classification.

While advances in light-sheet microscopy have extended imaging depth in organoids, maintaining high image quality throughout thick samples remains challenging. In practice, quantitative analyses are still largely restricted to organoids under roughly 100 $\mu m$ in diameter (*Dekkers et al., 2019*; *Ong et al., 2025*; *Zheng et al., 2023*), or hollow organoids with cavities, lumens, and epithelial structures (*de Medeiros et al., 2022*). These organoids are less optically opaque than larger organoids like gastruloids (*Beccari et al., 2018*; *Hashmi et al., 2022*), neuromuscular organoids (*Faustino Martins et al., 2020*), or cancer spheroids (*Zanoni et al., 2016*), which can reach diameters of 300 microns and more.

For large organoids, multiphoton microscopy provides a powerful alternative due to its ability to penetrate deep into thick tissues with minimal photodamage (*Schießl and Castrop, 2016*). This technique utilizes longer wavelengths of light to excite fluorescent molecules within the specimen, allowing the visualization of cellular structures and interactions in high resolution (*Helmchen and Denk, 2005*). Furthermore, it avoids drawbacks of confocal or light-sheet microscopy on large, densely packed samples, such as strong intensity gradients, image blurring, and reduced axial information due to light scattering, aberrations, degradation, or divergence of the light-sheet (*Hobson et al., 2022*). Finally,

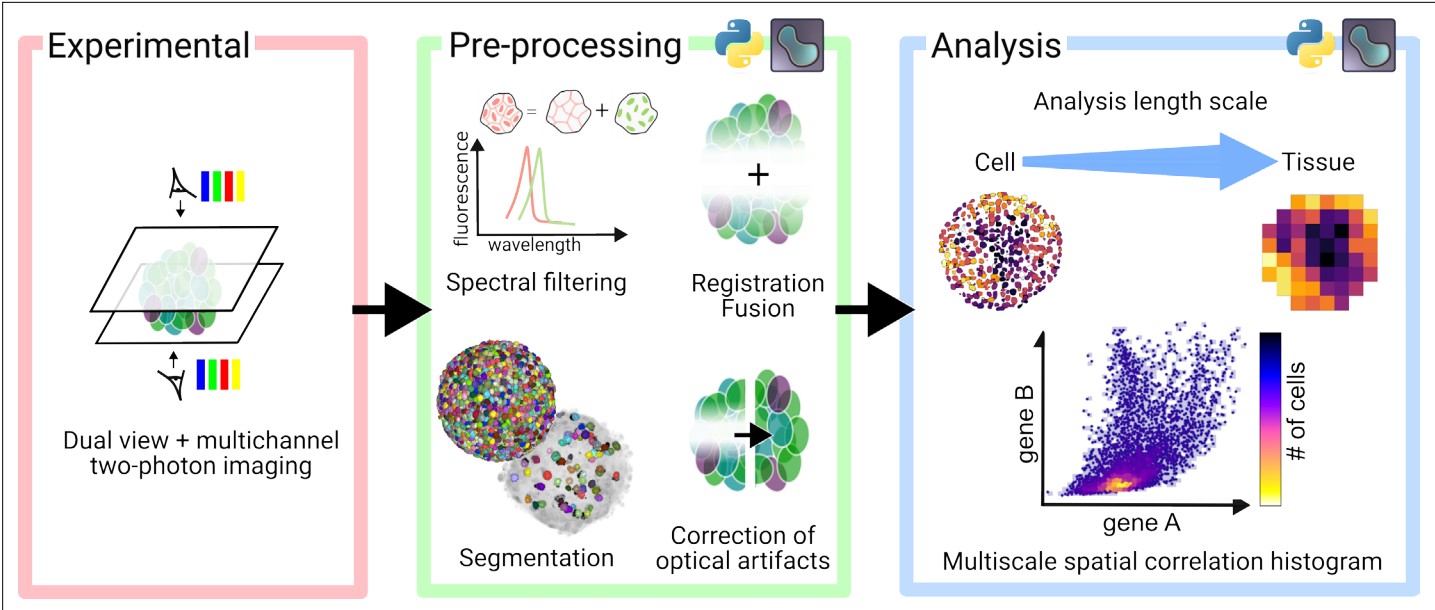

**Figure 1.** Overview of the experimental and computational pipeline for in toto imaging of organoids at cellular resolution and multiscale analyses.

two-photon microscopes are typically more accessible than light-sheet systems and allow for straight-forward sample mounting, as they rely on procedures comparable to standard confocal imaging.

## Gastruloids: A toy model for an integrated pipeline to analyze organoids development *in toto* at cell resolution

We have recently developed two-photon live imaging protocols to image developing gastruloids over multiple hours (*Hashmi et al., 2022*; *Gsell et al., 2023*). Gastruloids are mouse embryonic stem cells that self-organize into 3D embryonic organoids, sharing similarities with their in vivo counterparts (*Beccari et al., 2018*). Recent studies have shown that, within a few days, gastruloids undergo significant morphological changes, developing structures that closely resemble organs both genetically and morphologically, such as neural tube-, (*Veenvliet et al., 2020*) gut-, and cardiac-like structures (*Rossi et al., 2021*). Live two-photon imaging of gastruloids around their mid-plane has enabled us to propose biophysical models for their early symmetry breaking and elongation, using coarse-grained 2D analysis of collective tissue flows and gene patterning (*Gsell et al., 2023*). However, investigating in a full 3D context the relation between cellular fate and local tissue architecture during gastruloids morphogenesis (*Lenne and Tlili, 2023*) is still a key challenge. Indeed, it necessitates imaging in toto these dense 3D cell aggregates, which have a typical diameter above 200 microns and are highly light-diffusive objects.

In this work, we present an integrated pipeline to: (1) image multiple immunostained and cleared gastruloids in whole-mount configuration using two-photon microscopy, capturing both *in toto* and cellular scale details; (2) process and filter the resulting 3D images to correct optical artifacts from deep imaging and segment individual cell nuclei; and (3) analyze gene expression patterns, cellular events, and morphologies in 3D across multiple spatial scales (*Figure 1*).

## Results

### An experimental and computational pipeline for *in toto* organoid imaging

We have developed a versatile pipeline to image, process, and analyze tens of immunostained gastruloids (with diameters ranging from 100 to 500 $\mu m$). The experimental pipeline consists of sequential opposite-view multi-channel imaging of cleared samples, imaged with a commercial two-photon microscope (*Figure 1*). Afterwards, multiple processing steps are performed to extract meaningful information from the acquired images: (i) spectral unmixing to remove signal cross-talk, (ii) dual-view

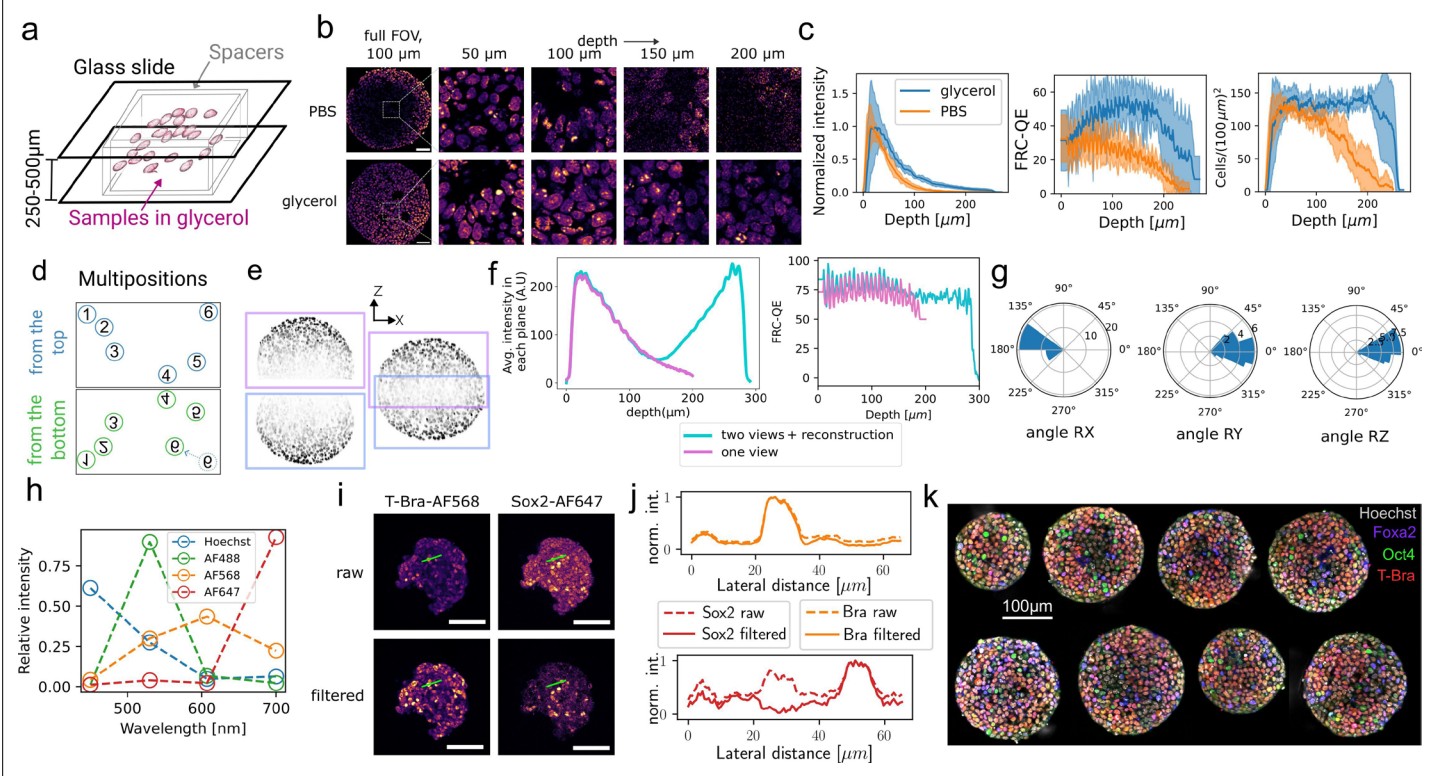

**Figure 2.** Dual-view multi-color two-photon imaging. (**a**) Multiple gastruloids are mounted in glycerol clearing medium in between two glass slides using spacers. (**b**) Mounting gastruloids in 80% glycerol significantly improves imaging at depth compared to phosphate-buffered saline (PBS) mounting. Insets show images at different depth in a central square region of 60×60 µm (**c**) Quantification of intensity, Fourier ring correlation quality estimate (FRC-QE) image quality metric, and density of segmented nuclei as a function of depth in the central region depicted in (**b**) for both glycerol- and PBS-mounted samples. Shaded areas show standard deviation for six (PBS) and nine (glycerol) gastruloids. (**d-f**) Top and bottom positions are assigned using a pattern-matching algorithm that is robust to residual movements of individual gastruloids, as illustrated for position 6 (**d**). Respective image stacks are registered using rigid 3D transformations and fused applying sigmoid fusion weights (**e**). This leads to *in toto* coverage of whole gastruloids, in which individual cells are still visible in the mid-plane (**e**), as shown for a gastruloid of typical size of 250 µm diameter. The average intensity per plane and FRC-QE are shown in (**f**) for one- and dual-view images. (**g**) Polar histogram plots of rotation angles around X-,Y-, and Z-axis from the 3D registration of dual-view top and bottom acquisitions, pooled for 37 aggregates. The sample slide was flipped around the X-axis. (**h–j**) Multi-color detection using two-photon excitation can lead to significant spectral cross-talk, as shown in the apparent emission spectra detected on gastruloids stained with a single fluorophore species (**h**). Using spectral unmixing, false-positive signal due to spectral cross-talk is effectively removed, as shown in the raw and filtered images (here at 50 µm depth) of gastruloids stained with T-Bra-AF568 and Sox2-AF647 (**i**) and the quantification of the intensity along a line (10 pixels width) across few nuclei for raw and spectrally filtered images (**j**). (**k**) Four-color images of the mid-plane of 77 hr gastruloids stained with Hoechst, FoxA2-AF488, Oct4-AF568, and T-Bra-AF647, obtained with dual-view four-channel two-photon imaging and spectral unmixing. The gastruloids were acquired in the same experiment. Scale bars are 50 µm in (**b, i**) and 100 µm in (**k**).

The online version of this article includes the following figure supplement(s) for figure 2:

**Figure supplement 1.** Benchmarking of dual-view two-photon imaging and spectral unmixing.

**Figure supplement 2.** Illustration of the sigmoid function that is used for weighted fusion of dual-view images, see *Equation 3*.

registration and fusion to reconstruct *in toto* images, (iii) sample and single cell segmentation, and (iv) signal normalization across depth and channels. Finally, reconstructed *in toto* images are analyzed at different scales, from cell-level correlations of different genes to coarse-grained maps of gene expression patterns, cell shapes, densities, and division events. All processing and analysis tools are open-source and Python-based. They are openly accessible as notebooks and interactive napari plugins.

## In toto multi-color two-photon imaging

To facilitate dual-view imaging, we mounted immunostained gastruloids between two glass coverslips using spacers of defined thickness (typically 250–500 µm, adapted to match the size of the gastruloids without compressing them), allowing us to image the sample iteratively from two opposing

sides (*Figure 2a*, see Methods Sample preparation Mounting). To optimize imaging performance, we compared several refractive index matching mounting mediums, glycerol (*Ahmad et al., 2021*), ProLong Gold Antifade mounting medium (Thermo Fisher), and optiprep (*Boothe et al., 2017*), on gastruloids labeled with Hoechst nuclei stain (see Methods Gastruloid culture). We identified 80% glycerol as the mounting medium with best clearing performance (*Figure 2b*, *Figure 2—figure supplement 1c*), leading to a threefold/eightfold reduction in intensity decay at 100 *μm*/ 200 *μm* depth compared to mounting in phosphate-buffered saline (PBS) and superior performance to gold antifade and live-cell compatible optiprep medium (*Figure 2—figure supplement 1c*). At these depths, information content was significantly improved by 1.5- and threefold, quantified using Fourier ring correlation quality estimate (FRC-QE) (*Preusser et al., 2021*; *Figure 2c*). Segmenting cell nuclei in images acquired on glycerol-cleared samples reliably detected cells at depth up to 200 *μm*, whereas a continuous decline in cell density was observed for PBS mounting, with four times fewer cells detected at 200 *μm* depth (*Figure 2c*, see details on the segmentation in the next section). These degradation effects were much more pronounced in confocal imaging even in glycerol clearing, which showed 2× lower intensity and 8× lower FRC-QE than two-photon imaging at 100 *μm* depth (*Figure 2—figure supplement 1a and b*).

Nevertheless, two-photon microscopy in glycerol beyond 200 *μm* depth suffered from decreased signal-to-noise-ratio (SNR) (*Figure 2b and c*). To properly image larger gastruloids *in toto*, a second detection side was required. We, therefore, flipped the sample slide and re-imaged the same set of gastruloids from the opposing side. Whereas the flipping was done manually, all mounted gastruloids were imaged automatically after defining their positions in the imaging software. A typical acquisition took about 5–10 min per gastruloid per side for 1 *μm* z-spacing and full field-of-view (318 *μm* with 0.62 *μm* pixel size at 40× magnification). To reconstruct *in toto* image stacks, we automatically identified opposite views of the same gastruloid by applying a pattern matching algorithm to gastruloids positions in the two acquisitions (*Figure 2d*, see Methods Preprocessing: Dual-view registration and fusion). Next, we registered each pair of image stacks acquired on the same aggregate using a rigid 3D transformation determined by a content-based block-matching registration algorithm (*Ourselin et al., 2000*) previously applied for multi-view and time registration of light-sheet imaging data (*McDole et al., 2018*; *Guignard et al., 2020*). Finally, the registered image stacks were fused using a sigmoid decay function to weight contributions of the two sides (*Figure 2e,f*, *Figure 2—figure supplement 1*). The registration was performed on the ubiquitous nuclei channel, i.e., Hoechst staining. In some instances, flipping of the slide resulted in additional minor rotations (on average (−13±11)°, (−13±18)° and (−5±10)° around the X-, Y-, and Z-axis for n=37 aggregates), which occasionally prohibited convergence of the registration algorithm (*Figure 2g*). For these cases, we implemented a napari plugin to pre-register the images manually and visualize the result in 3D (Figure 6). Fused gastruloid images exhibit about fourfold lower intensity in the mid-plane compared to the outer ends, but approximately constant FRC-QE across depth (*Figure 2f*).

A common limitation in fluorescence imaging is the low number of targets that can be excited simultaneously and spectrally discriminated. To image multiple targets in parallel, we excited four fluorophore species (Hoechst, AF488-, AF568-, AF647-tagged secondary antibodies on immunostained gastruloids) simultaneously. Thanks to the broad two-photon excitation spectra, the four dye species could be excited with just two laser lines, 920 nm and 1040 nm. However, detecting their emission simultaneously on four non-descanned detectors resulted in significant signal cross-talk, due to spectral overlap (*Figure 2h–j*). We circumvented this problem by applying spectral decomposition (*Dunsing et al., 2021*). We calibrated spectral patterns, i.e., apparent emission spectra for each separate fluorophore species, in gastruloids stained with a single fluorophore species, and used these as reference patterns to decompose multi-color data (*Figure 2h*). Because of chromatic effects due to dispersion, spectral patterns were not constant at different depths in the sample. For example, shorter wavelength emission decays stronger with imaging depth than longer wavelength fluorescence (*Figure 5—figure supplement 1b*). To account for this, we performed the unmixing with depth-dependent spectral patterns (see Methods Preprocessing: Spectral unmixing). Thereby, we could significantly remove cross-talk, for example, false positive cells in the far-red detection channel that resulted from cross-talk of bright cells visible in the red channel (*Figure 2i and j*). Notably, spectral decomposition with four-channel detection was even possible for spectrally strongly overlapping fluorophores, which we verified on gastruloids with transgenic H2B-GFP expression in the nuclei, stained

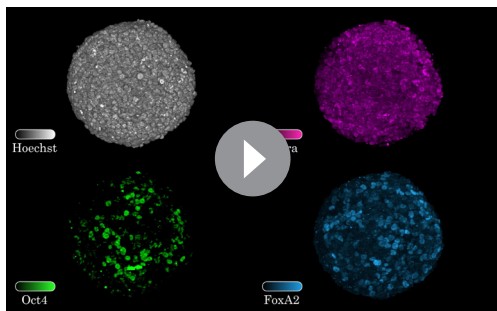

**Video 1.** 3D rendering of a 78 hr small sample processed through the pipeline. Hoechst is in gray, T-Bra in magenta, Oct4 in green, and FoxA2 in blue.
https://elifesciences.org/articles/107154/figures#video1

with E-cadherin (Ecad)-AF488 at cell membranes (*Figure 2—figure supplement 1e and f*). We thus applied spectral unmixing to all data presented throughout this work. This way, we could generate cross-talk-free four-color images of several biologically relevant markers imaged simultaneously in multiple gastruloids in the same experiment (*Figure 2k*, *Videos 1–3*).

**Video 2.** Z-stack of a 78 hr big sample processed through the pipeline. Hoechst in gray, T-Bra in red, Oct4 in green, and FoxA2 in blue.
https://elifesciences.org/articles/107154/figures#video2

## Single cell segmentation

Quantifying cell-scale gene expression and nuclei morphological properties requires accurate instance segmentation of individual nuclei. In early-stage gastruloid datasets acquired with two-photon microscopy, accurate segmentation of nuclei is hindered by high nuclei packing (*Figure 2b*, *Figure 3—figure supplement 1*), shape heterogeneity, and spatially varying SNR due to optical aberrations and scattering, especially deeper in the sample. We qualitatively assessed the performance of two open-source state-of-the-art methods for 3D segmentation of nuclei in dense environments, i.e., Cellos (*Mukashyaka et al., 2023*) and AnyStar (*Dey et al., 2024*), and observed unsatisfying results (*Figure 3—figure supplement 2a*). We also considered 3DCellScope (*Ong et al., 2025*), but its closed-source nature and required a paid licence to access full segmentation features. We thus trained a custom Stardist3D (*Weigert et al., 2020*) model to segment nuclei, as it excels with separating star-convex objects in close contact.

A mid-plane section of the StarDist prediction is shown *Figure 3a*, exhibiting a qualitatively good result on outer layers as well as inner part of the sample. To increase robustness, the network was trained on three datasets acquired with two-photon microscopy in different experimental conditions, namely on (1) live-imaging of Histone 2B (H2B)-GFP gastruloids, (2) live-imaging of sparsely labeled mosaic gastruloids (composed of a mix of 75% of non-fluorescently labeled cells and 25% of cells expressing H2B-GFP under the Brachyury gene promoter *Hashmi et al., 2022*), and (3) 96 hr fixed samples stained with Hoechst. This combined dataset presented variations in labeled nuclei density, nuclei texture, and in voxel dimensions (*Figure 3b*, see Table 3 and *Figure 3—figure supplement 1a*). In total, we annotated 4414 nuclei in 3D and used data augmentation (see Methods Nuclei segmentation) to enrich the training datasets (*Figure 3b*). Both during training and inference, we applied local contrast enhancement and normalization to all datasets by clipping intensity histograms

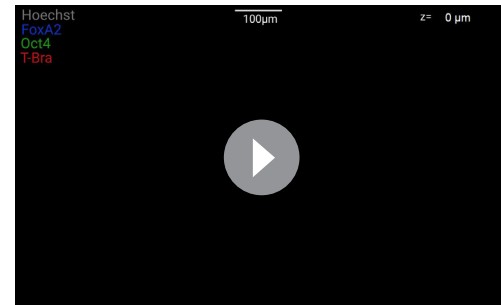

**Video 3.** Z-stacks of eight samples imaged at 78 hr, all in the same acquisition, and processed through the pipeline. Hoechst in gray, T-Bra in red, Oct4 in green, and FoxA2 in blue. Depth in the sample is indicated in the bottom right corner.
https://elifesciences.org/articles/107154/figures#video3

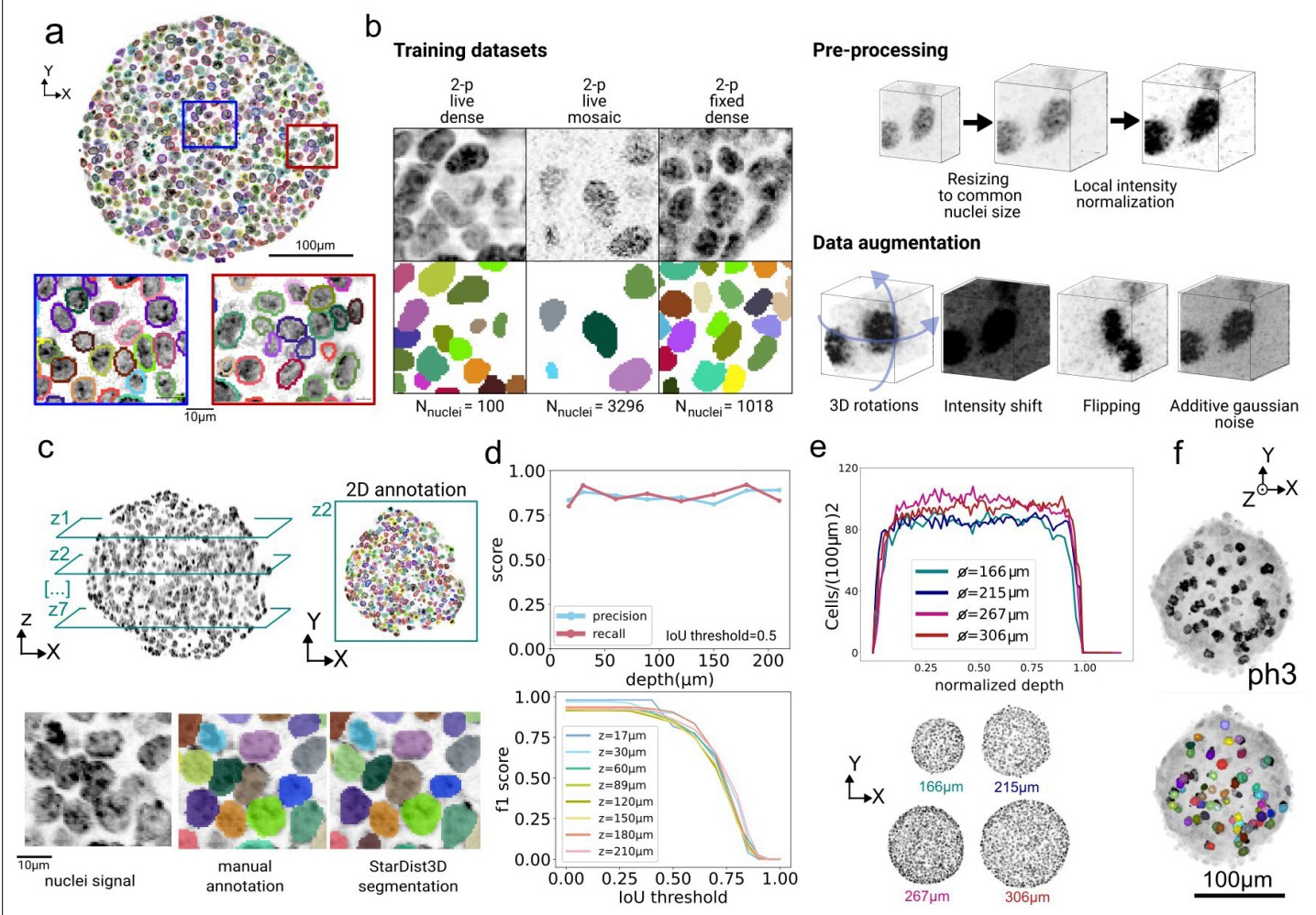

**Figure 3.** 3D nuclei segmentation with StarDist3D. (**a**) Exemplary segmentation (contours) of Hoechst-stained nuclei (gray) using custom-trained StarDist3D model. Insets show segmentation results in inner and outer cell layers, at 150 $\mu m$ depth. (**b**) Three different datasets were annotated and used for training. Images show exemplary annotated patches (top row: intensity images, bottom row: label images). Before training, images were resampled to a common voxel size of 0.62×0.62×0.62 $\mu m^3$. Image contrast when enhanced using local histogram clipping and normalization. During training, data augmentation was used to maximize training input. (**c**) To evaluate segmentation performance, entire z-planes of a gastruloid were annotated at different depths in 2D, on a 250 $\mu m$ size gastruloid imaged from two views and fused, and the segmentation result was compared with the ground truth. (**d**) Top: Precision and Recall at IoU threshold 0.5 as a function of depth. Bottom: F1 score as a function of IoU thresholds, color represents depth. (**e**) Cell density obtained from 3D segmentation of gastruloids of different diameters. The corresponding images at the mid-plane are shown at the bottom. (**f**) 3D rendering of the detection of mitotic cells using StarDist3D. Top image shows phospho-histone H3 (ph3)-stained sample, bottom image shows detected cell divisions overlayed with the ph3 signal.

The online version of this article includes the following figure supplement(s) for figure 3:

**Figure supplement 1.** Quantification of StarDist3D performance.

**Figure supplement 2.** Qualitative benchmark of our segmentation model.

**Figure supplement 3.** Validation of the Cellpose–SAM pretrained model for nuclear segmentation and morphology in gastruloids.

computed in local boxes at the first and 99th percentile and by mapping intensities between 0 and 1. This step was crucial to homogenize the intensity distribution across and inside datasets during training, and improved segmentation quality by reducing the number of false detections at depths at which the SNR is reduced.

To assess the performance of the training, we manually annotated 2D planes at different depths throughout a sample (see *Figure 3—figure supplement 1c*, sample 1) and compared the result of StarDist3D to the ground truth, both in terms of quality of nuclei detection and volume reconstruction (*Figure 3c*). To evaluate the performance, we computed the recall and the precision (resp. quantifying

how many relevant items are retrieved and how many retrieved items are relevant), and the F1 score as the harmonic average between the two. We show in *Figure 3d* that our trained model recovers a constant precision and recall across the full depth, and the F1 score is superior to 0.8 up to intersection over union (IoU) threshold 0.5. Comparing the recall and precision to the ones obtained with global contrast enhancement, i.e., histogram clipping based on the percentiles of the whole image, our local contrast enhancement scheme mitigates both false positives and false negatives in deeper z planes (*Figure 3—figure supplement 1d–e*). It is still necessary to image from two views to recover all the nuclei, as shown *Figure 3—figure supplement 1f-g*, even using the local contrast enhancement method. Our score measurement is based on computing the IoU of 3D objects based on 2D images. As manually annotating many dense organoid datasets for validation is a slow and cumbersome process, we designed a validation method based on comparing the 3D prediction of StarDist3D with 2D annotations in several XY planes along the z-axis. We validate this approach on *Figure 3—figure supplement 1b* using four different samples in which we chose XY planes spaced evenly along the z-axis, and on which nuclei were annotated in 3D. The F1 score obtained in 3D (3D prediction compared with the 3D ground-truth) is well approximated by the F1 score obtained in 2D (2D predictions compared with the 2D sliced annotated segments). The difference between the 2 scores was at most 5%. Finally, we observed that the model preserves a constant cell density across depth in spherical gastruloids of up to 475 $\mu m$ in diameter (*Figure 3e*).

Overall, our custom model achieved an F1 score of 85±3% at 50% IoU. Using a variety of annotated datasets, dual-view imaging and a local contrast enhancement algorithm, we trained a powerful StarDist3D model that shows robust segmentation performances regardless of depth, sample size, and experimental conditions.

## Maps of proliferation and cell density

Differential division and growth rates in tissues are key drivers of morphogenetic changes (*Fox et al., 2018*). Therefore, quantifying tissue-scale gradients in cell proliferation is crucial for understanding the morphogenesis of gastruloids. To detect division events, we stained gastruloids with phosphohistone H3 (ph3) and trained a separate custom Stardist3D model using 3D annotations of nuclei expressing ph3 (see Methods Mitotic cells segmentation). This model together allowed us to detect nearly all mitotic nuclei in whole-mount samples for any stage and size (*Figure 3f* and *Video 4*), and we used minimal manual curation to correct remaining errors. To probe quantities related to the tissue structure at multiple scales, we smooth their signal with a Gaussian kernel of width σ, with σ defined as the spatial scale of interest. From the segmented nuclei instances, we compute 3D fields of cell density (number of cells per unit volume), nuclear volume fraction (ratio of space occupied by nuclear volume within the local averaging volume, as defined in the Methods Masked Gaussian convolution to probe spatial fields at different scales with a scale of σ), and nuclear volume at multiple scales. We plot each field on *Figure 4a* at three spatial scales ranging from the nuclei scale, i.e., σ set to the average nuclear radius, to the tissue scale, i.e., σ set to six times the average nuclei radius. Taking this smoothing approach, we computed fields of division density (number of division events per unit volume), that represent the local density of cells in mitosis, from the division labels detected on ph3 as described above. An increased volumetric division density can result from either elevated cell proliferation or higher cellular volumetric density. To distinguish between these effects, we computed a proliferation probability field that represents the local fraction of cells undergoing mitosis within the 3D sample (by dividing the division density field by the cell density field). A cross-section of each of these 3D maps is represented in *Figure 4b*, showing

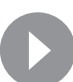

150

**Video 4.** Z-stacks of two samples stained with ph3, on the right at 72 hr and on the left at 96 hr. Detected divisions are visualized as multicolor ROIs superimposed with the image, where only the contour is shown.
https://elifesciences.org/articles/107154/figures#video4

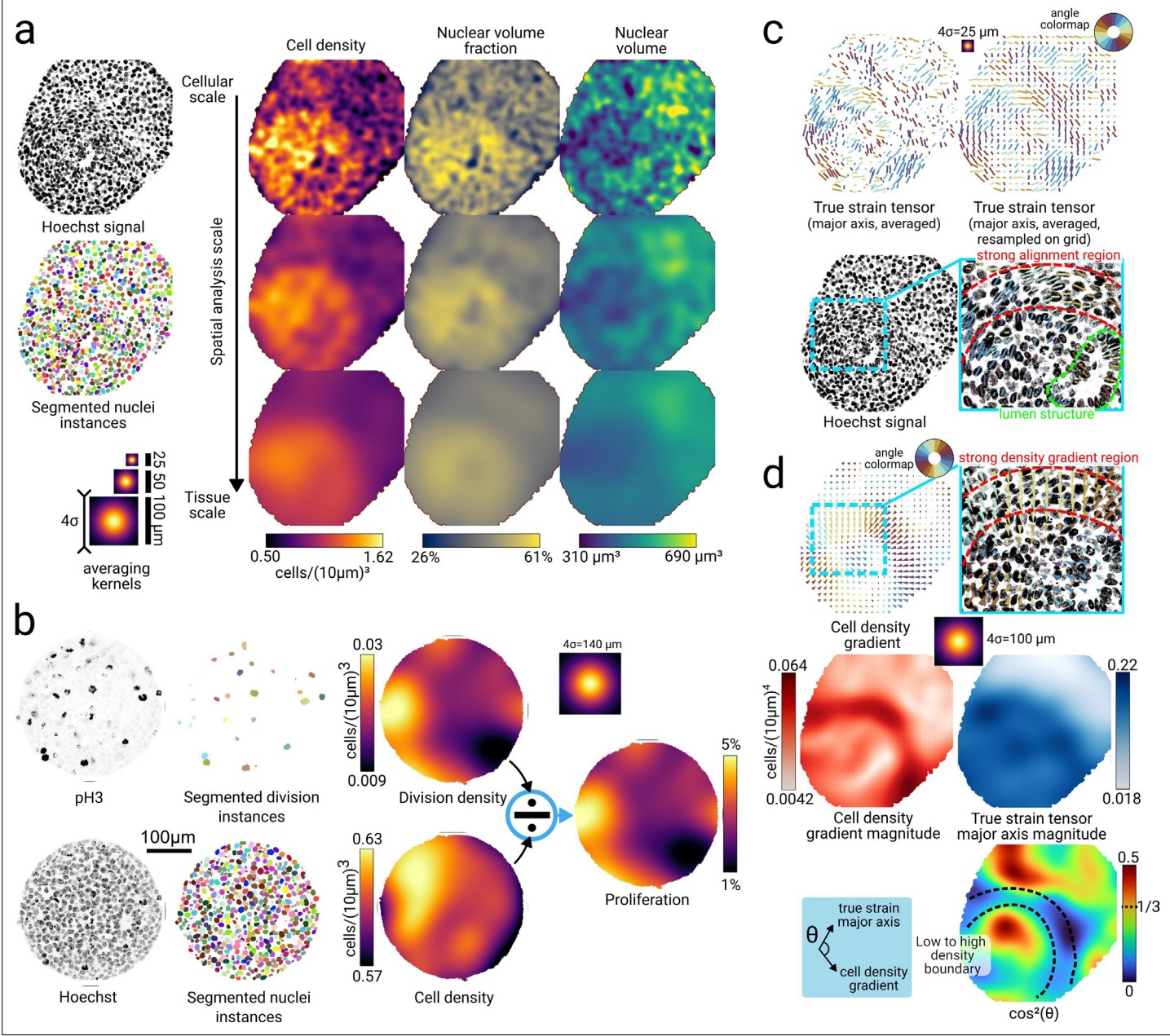

**Figure 4.** Cell-to-tissue scale morphological analysis. (**a**) Segmenting individual nuclei instances in 3D allows a complete characterization of object packing by quantifying cell density, nuclear volume fraction, and nuclear volume. Each quantity is shown at three resolutions ranging from the cell scale to the tissue scale, which correspond to convolution with Gaussian averaging kernels with σ=6, 12 and 25 $\mu m$, respectively. (**b**) Divisions stained with ph3 and nuclei stained with Hoechst are segmented using StardDist3D and used to compute respectively maps of division density and cell density. The proliferation is defined as the ratio of the division density with the cell density, showing the local fraction of cells undergoing mitosis. (**c**) Cell-scale fields of deformation, quantified by the major axis of the nuclei true strain tensors. A version sampled on a regular grid is provided for better readability. The deformation field reveals a region of spatially persistent alignment that overlaps with the density boundary and highlights nuclei forming the lining of a lumen. (**d**) We compute the tissue-scale field of squared cosine of the angle between the cell density gradient and the true strain tensor major axis to quantify local regions of anti-alignment. The vector magnitudes of the two fields are shown to add perspective to the analysis in regions where the gradient or the local deformation is small. All quantities shown in this figure are computed in 3D, but we show a single z-slice (vector quantities are projected onto the slice) for convenience.

The online version of this article includes the following figure supplement(s) for figure 4:

**Figure supplement 1.** Reproducibility of the high to low cell density boundary.

a typical heterogeneous pattern of division density and cell density. These maps can be visualized in parallel for several samples, which allows systematic comparison of patterns for high-throughput acquisitions. Analyzing proliferation fields in 3D gives us insight on growth patterns during development and is key to understand and quantify gastruloid morphogenesis. Importantly, this quantification can be applied across various developmental stages, providing quantitative insights into growth phases under different stages or media conditions. A proliferation pattern for a strongly elongated 120 hr gastruloid grown in a modified culture protocol is shown in *Figure 4—figure supplement 1b*.

## Morphometric analysis

Tissue-scale patterns of cell/nuclei deformations in developing embryos or organoids are good indicators of the existence of mechanical stresses which can contribute to tissue morphogenesis (*Stooke-Vaughan and Campàs, 2018*) and affect in return cell differentiation (*Chan et al., 2017*). To explore changes in tissue structure and organization during gastruloid development, we studied spatial heterogeneities in cell densities and cell deformations emerging during the first morphogenetic phase of gastruloids development, which consists in a global tissue elongation between 72 and 96 hr after their initial aggregation. We have shown in previous work (*Hashmi et al., 2022*; *Gsell et al., 2023*) that such elongation is associated with the polarization of the gastruloid, exhibiting a posterior pole rich in E-cadherin and constituted of a mixed population of both endoderm and nascent mesoderm tissues, and an anterior pole poor in E-cadherin constituted of cardiac and vascular mesoderm tissues which are mesenchymal. Here, to prevent any relaxation of cellular deformation caused by tissue fixation, we performed 3D acquisition of live gastruloids with an H2B-GFP marker. We illustrate the full approach on a single sample in *Figure 4c–d*, and show that the findings are reproducible across different samples and views (see *Figure 4—figure supplement 1a*). The cell density field revealed a high level of heterogeneity at the nuclei scale, but analysis at different scales always showed the presence of two tissue-scale populations of low and high cell densities located, respectively, in the posterior and anterior part of the sample, separated by a sharp boundary characterized by the length scale of a few nuclei. These two populations and the boundary were also visible, although less clearly, from the nuclear volume and nuclear volume fraction fields, with the high-density population showing a higher nuclear volume fraction but lower nuclear volumes. The nuclear volume fraction indicates regions of high packing located in the high-density population, with a maximum of 61%, close to the theoretical value for random close packing of spheres (64%). As extra-cellular space volume fraction is low in such sample (*Gsell et al., 2023*), this indicates high nuclei-to-cell volume ratios, which suggests that nuclei shapes are a good proxy for cell shapes, and thus nuclei could be relevant in cell-scale mechanotransduction processes.

To quantify cell deformation, we first computed the inertia tensor from each segmented nuclei instance and obtained the true strain tensor (*Tlili et al., 2015*) for each cell (see Methods Morphometric analysis), which is a size-independent measure of object deformation. This justifies averaging these tensors among nuclei with large size variations by effectively removing the size bias. We plot the major axis (associated with the largest eigenvalue) of the true strain tensor for each nucleus on *Figure 4* at the nuclei scale. The major axis is oriented along the direction of the largest deformation of the nuclei. Close to the low-to-high density boundary, we observed that nuclei are elongated in the direction of the boundary, i.e., perpendicular to the gradient of cell density. We further quantified this observation by computing the angle $\theta$ between the gradient of cell density and the major axis of the true strain tensors. In *Figure 4d*, we plot the continuous field representing the quantity $\cos^2(\theta)$ averaged at the tissue scale. For random orientations of the major axis with respect to the gradient of cell density, the average value of $\cos^2(\theta)$ is 1/3, and the value approaches 0 or 1 for respectively perpendicular and parallel orientations. The field reveals a wide region with values below 0.1 around the boundary, confirming that nuclei are elongated along the tissue boundary. Other regions show comparatively smaller values of the local gradient amplitude or no tissue-scale alignment. These descriptions appear reproducibly in several samples, see *Figure 4—figure supplement 1a*, and are consistent with an interpretation of the boundary as a region where mechanical stresses are generated between gastruloids anterior and posterior poles. This analysis also shows how nuclei shapes at small scales are appropriate proxies for deformation inside and around tissue micro-structures like lumen and cavities, which allows for an accurate description of the local epithelial architecture. At the

tissue scale, nuclei shapes can serve as a proxy for tissue mechanical stresses in the context of tissues modeled as visco-elastic liquids with residual elasticity in cell deformation (*Tlili et al., 2020*).

## Quantitative analysis of gene expression at global and cell-scale level

A key interest in *in toto* imaging is that it allows quantifying expression gradients of genetic markers and proteins of interest across entire organoids. In the case of gastruloids, gradients of differentiation emerge during Wnt activation, prior or concomitant with symmetry breaking (*Hashmi et al., 2022*; *Suppinger et al., 2023*; *Villaronga Luque et al., 2023*; *van den Brink et al., 2014*). Due to the previously mentioned limitations of light-sheet microscopy for *in toto* coverage and quantitative intensity analyses for gastruloids beyond 100−200 *µm* diameter, investigations of such gradients have thus far been restricted to analyses of low resolution 2D images (*Villaronga Luque et al., 2023*; *Suppinger et al., 2023*). Because of this limitation, it is still unknown when gradients of differentiation are established during gastruloid development. To explore this, we analyzed images of gastruloids fixed at 77 hr, labeled with Hoechst, and additionally immunostained with T-Bra, a mesendoderm marker expressed upon Wnt-mediated cell differentiation. Previous investigations with 2D imaging indicated a gradient of higher T-Bra expression in outer cell layers and lower expression in the interior (*Hashmi et al., 2022*). Although our dual-view imaging pipeline delivers *in toto* coverage, images of the ubiquitous Hoechst nuclei stain still exhibited intensity gradients along the z direction and between outer and inner cell layers (*Figure 5a*). We attributed these gradients to optically induced heterogeneities and developed a correction method that re-normalizes the channels of interest based on the local nuclei intensity. To this aim, we first corrected the intensity decrease, which is dependent on the emission wavelength and the depth (see *Figure 5—figure supplement 1*). We applied on this corrected signal a convolution with a 3D Gaussian kernel masked with the nuclei segmentation. The result of that convolution is a coarse-grained map of nuclei intensity, that we use to locally normalize the original image (*Figure 5a*, *Figure 5—figure supplement 2*). Since this normalization was done in 3D, the intensity values in the mid-plane of a gastruloid were enhanced while the intensities in the first and last z-planes remained constant (*Figure 5—figure supplement 3b*). Furthermore, the method effectively and robustly homogenized nuclei intensities across gastruloids from different acquisitions and imaging conditions (*Figure 5—figure supplement 3c and d*). We evaluated different sizes for the Gaussian kernel of the convolution and chose an optimal one, with a typical value around the diameter of the nuclei (10−15 *µm*), that lead to the most homogeneous Hoechst signal after re-normalization (see Methods Preprocessing: Intensity normalization and *Figure 5—figure supplement 3e*). This normalization scheme accounts for heterogeneities in gastruloids of arbitrary shape, as well as local cell density, and thus variations in the optical paths of excitation and fluorescence light rays that cannot be captured by a simple geometrical model.

When we applied the correction method to the T-Bra signal and evaluated re-normalized T-Bra intensity in a lateral section across the mid-plane of a 77 hr gastruloid, we still observed a gradient with higher intensity in outer cell layers (*Figure 5a*), while the nuclei signal recovered a flat profile. Besides reflecting a true T-Bra expression gradient, a possible explanation for this observation could be hindered penetration of primary and secondary antibodies being considerably larger than Hoechst stain. To evaluate this, we compared immunostained T-Bra signal with the intensity of a T-Bra reporter (H2B-GFP expressed under a T-Bra promoter) in gastruloids grown from a transgenic cell line fixed at 72 hr. We found a strong correlation between the two channels at all depths, excluding penetration artifacts (*Figure 5—figure supplement 4*).

In addition to optical artifacts, our re-normalization method also corrects for intensity variations effectively induced by residual rotations of aggregates after flipping, which tilt the optical axis with respect to the axis of the first acquisition (*Figure 2g*). This is illustrated in *Figure 5b*, providing an example where the nuclei signal exhibits an asymmetrical gradient due to the fusion of two opposite views that were subject to residual rotations. After re-normalization, the nuclei profile is homogeneous, whereas the profile in T-Bra drastically changed, showing a polarization in the opposite direction with respect to the initial gradient that was also visible both in the nuclei stain and T-Bra channel. This emphasizes the importance of our method to properly quantify gene expression gradients in 3D. The observed polarized T-Bra appearance is in agreement with our previous study, in which we have reported that T-Bra expression already commences to be polarized at 77 hr in some gastruloids (*Hashmi et al., 2022*).

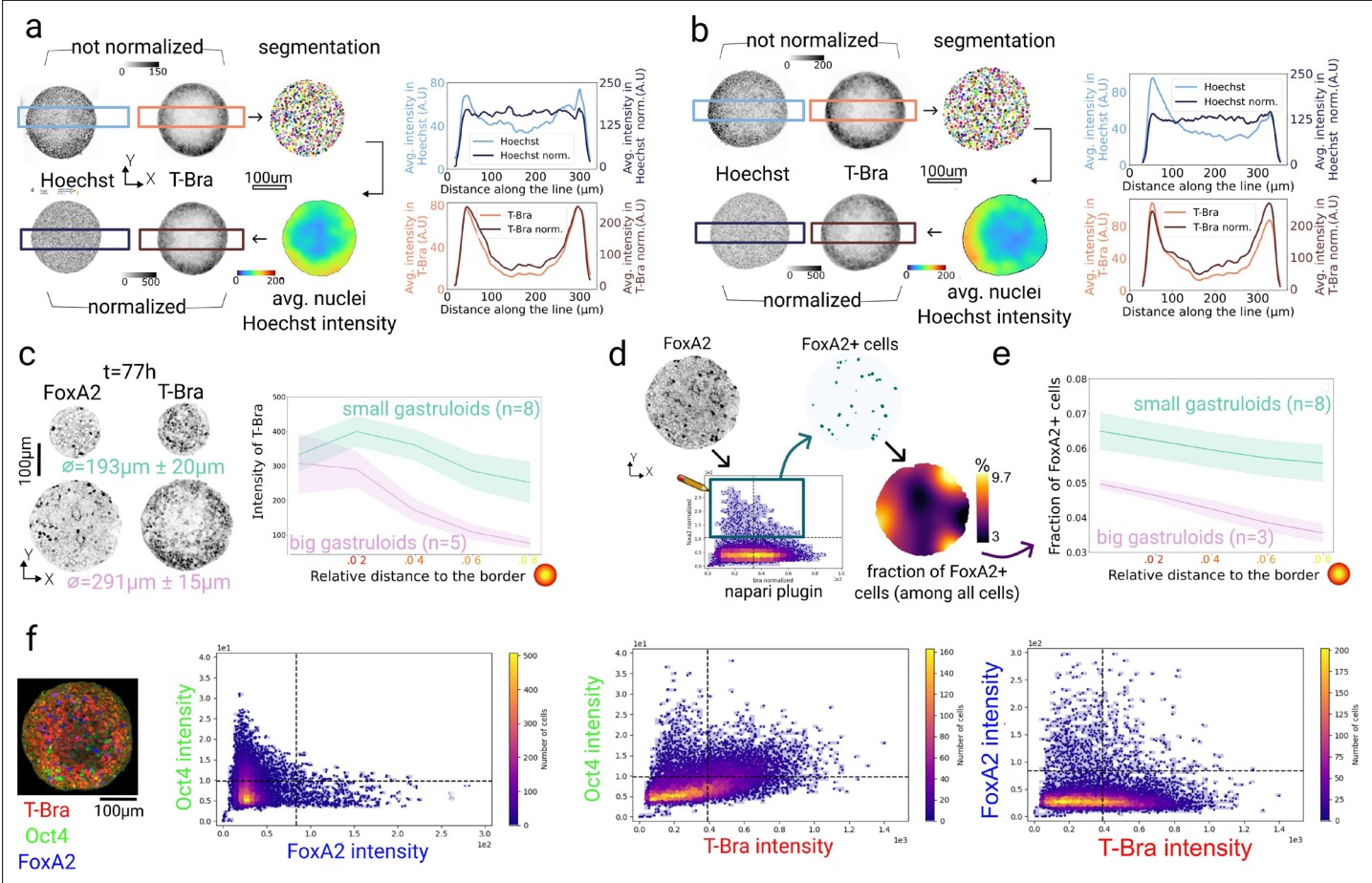

**Figure 5.** Coarse-grained and cell-level gene expression analysis. (**a, b**) Images of Hoechst and T-Bra in the mid-plane of 77 hr gastruloids. Using the nuclei segmentation, maps of averaged nuclei intensity were generated and used to locally re-normalize the T-Bra signal. The graphs on the right side of each panel show the intensity across rectangular sections of the mid-plane, shown in the images as blue/ brown boxes, before and after intensity normalization. This intensity profile is smoothed using a Gaussian kernel on the images to extract the large-scale intensity variations. (**a**) example of a radially symmetric gradient of nuclei intensity, from the exterior to the interior. (**b**) Example of an asymmetric gradient, the left side of the sample has a high nuclei intensity that is corrected after normalization, changing the T-Bra profile (bottom right graph). (**c**) Images of 77 hr gastruloids (mid-plane) of different size, immunostained for T-Bra and FoxA2 gene expression markers. From the re-normalized 3D images, radial intensity plots were generated for big (291±15 $\mu m$ average diameter, n=5) and small (193±20 $\mu m$ average diameter, n=8) gastruloids as a function of distance to the border. Shaded regions show standard deviations. (**d**) For one of the previously described samples fixed at 77 hr, FoxA2 positive cells are visualized from the re-normalized data using the interactive napari tools and the image thresholded accordingly. Using the binary image of FoxA2-positive cells, a map of local fraction of FoxA2-positive cells, rescaled by the local cell density, was computed, showing a radial pattern. (**e**) Analysis of the radial distribution, i.e., fraction of FoxA2-positive cells as a function of distance to the border, for big (278±16 $\mu m$ average diameter, n=3) and small (193±20 $\mu m$ average diameter, n=8) aggregates, computed from the coarse-grained maps as shown in (**d**). Shaded regions show standard deviations. (**f**) Three-color image of a 77 hr gastruloid immunostained for T-Bra, Oct4, and FoxA2. The three plots show a cell-level correlation analysis of the three markers for the entire gastruloid. Each dot in the correlation plot represents an individual cell. Shown are all three marker combinations. Quadrants show regions of background signal (-) and actual expression (+) for each marker in the respective color, defined by Otsu thresholding.

The online version of this article includes the following figure supplement(s) for figure 5:

**Figure supplement 1.** The wavelength-dependent normalization method.

**Figure supplement 2.** Illustration of the intensity model and of the normalization procedure based on a ubiquitous signal.

**Figure supplement 3.** Effect of spectral filtering and local intensity normalization on 3D gene expression analysis.

**Figure supplement 4.** Direct comparison of depth-dependent correlation of intensity between immunostained T-Bra signal and the intensity signal of a T-Bra reporter (H2B-GFP expressed under a T-Bra promoter) in gastruloids grown from a transgenic cell line fixed at 72 hr.

Because the intensity decay in depth is wavelength-dependent (*Figure 5—figure supplement 1b*), normalizing the different channels only based on the Hoechst intensity in the blue channel would have induced over-corrections in depth. We applied a wavelength-dependent correction to the different channels, by first measuring the characteristic length of exponential decay for the four colors independently (*Figure 5—figure supplement 1a* and c), and second by computing a map of Hoechst intensity corrected by this characteristic length. This gave us the equivalent of coarse-grained maps of nuclei intensity in each color, serving as a reference to correct the different channels (details in Methods Preprocessing: Intensity normalization).

We then applied the correction scheme to four-color 3D acquisitions of multiple gastruloids of different sizes (obtained by varying the initial cell number of the culture) imaged in the same slide, immunostained with T-Bra, the pluripotency marker Oct4, and the early endoderm marker FoxA2 (*Figure 5c*). We sorted these gastruloids into two groups and observed an approximately constant T-Bra expression for small aggregates (193±20 *µm* average diameter, n=8), whereas big aggregates (291±15 *µm* average diameter, n=5) showed a clear decay of T-Bra expression towards the interior (*Figure 5c*). Notably, we quantified T-Bra intensity as a function of distance to the gastruloid border using the *in toto* 3D image stack, thus reflecting expression gradients in 3D. For FoxA2, a similar pattern was observed, but expression appeared in much fewer cells (*Figure 5d*). Because FoxA2 signal was sparse and noisy, we could not compute simply the map of expression as done previously for T-Bra. Instead, we computed a map of the fraction of FoxA2 cells in a given neighborhood. We first thresholded the signal in segmented nuclei, using the interactive visualization tools developed in napari (see *Figure 6* and Methods). This allowed us to identify the correct intensity threshold and generate a binary mask of FoxA2 expression. We then computed the number of FoxA2-positive cells rescaled by the local nuclei density, using the same method as applied for proliferation analysis, except that we consider FoxA2 positive cells instead of ph3-positive cells. The coarse-grained map of FoxA2-positive cells was then analyzed radially for gastruloids of various sizes, small gastruloids of 193±20 *µm* average diameter (n=8) and bigger gastruloids of 278±16 *µm* average diameter (n=3). We observed slightly higher FoxA2 levels in outermost cell layers, and in general more cells expressing FoxA2 in small gastruloids proportionally to the total number of cells. We applied the same method of radial analysis on 3D maps of Sox2 fraction, see *Figure 5—figure supplement 3a* on five gastruloids at 60 hr. Cells expressing Sox2 are heterogeneously distributed but there is no radial gradient. Taken together, these analyses demonstrate that coarse-grained maps coupled with radial profiling can quantify large-scale patterns and gradients of gene expression across gastruloids.

Our ability to image multiple expression markers in the same acquisition without cross-talk and to reliably segment all cells allows correlating expression of different genes at the single cell level. To demonstrate this, we analyzed correlations between T-Bra, Oct4, and FoxA2 in the data described above, after spectral filtering and intensity re-normalization. Both processing steps were crucial to obtain unbiased results (*Figure 5—figure supplement 3f and g*). To discriminate marker positive from cells only showing background levels, we applied Otsu thresholding on the cells' intensity distribution for each marker and confirmed the distinction by visible inspection.

We expected that FoxA2 and Oct4 are mutually exclusive (since cells are either in the process of endoderm differentiation or still pluripotent). A similar behavior was expected for T-Bra and Oct4/T Bra and FoxA2, although some co-expression could be present since T-Bra marks very early mesendoderm differentiation en route to meso- or endoderm and Oct4 is a relatively long-lived pluripotency marker (*Abranches et al., 2013*).

We indeed observed that Oct4 and FoxA2 appeared as two mutually exclusive populations (*Figure 5f*, left). On the contrary, while the brightest Oct4 cells also showed low T-Bra signal and vice versa, a large pool of T-Bra and Oct4 positive cells appeared, not showing a striking correlation pattern for most cells (*Figure 5f*, center). For FoxA2 and T-Bra, a clear anti-correlation was observed, with cells appearing in all four quadrants (*Figure 5f*, right). Interestingly, we overall detected fewer FoxA2-positive cells than T-Bra-positive cells. While we found many cells that were T-Bra positive and FoxA2 negative, most of the FoxA2 positive cells showed at least some T-Bra signal. This indicates that T-Bra expression might be required to activate FoxA2. However, cells expressing both markers still showed a negative correlation, suggesting that once FoxA2 is activated, T-Bra expression is repressed. Overall, these data suggest a sequence of cellular transitions from Oct4 progressively to T-Bra, from which a fraction of cells, presumably the ones expressing low T-Bra levels, will commit to

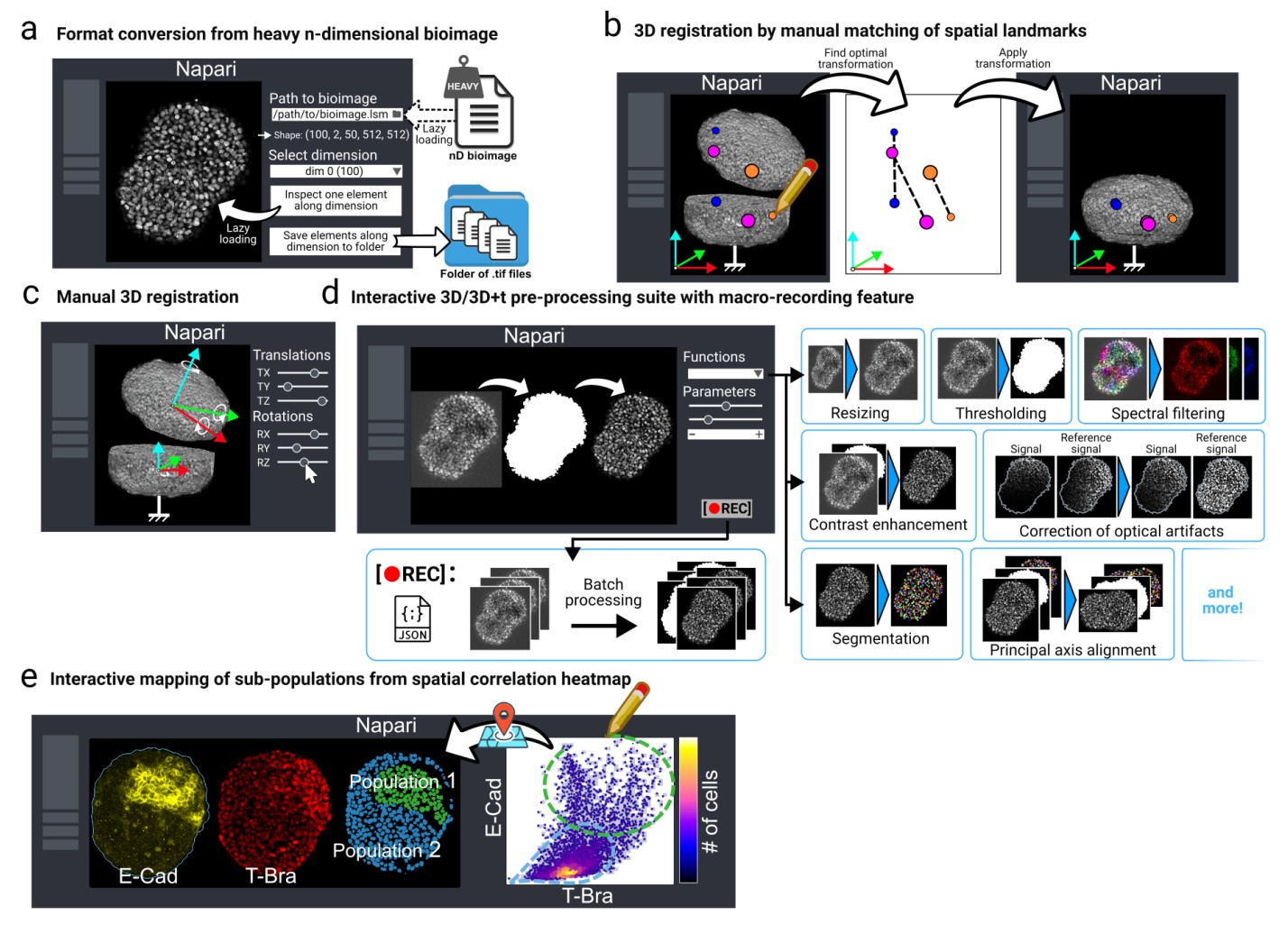

**Figure 6.** Interactive napari plugins for bioimage handling, manual registration, preprocessing, data exploration, and analysis. (**a**) A napari plugin for exploring and processing large nD bioimages. Users can load datasets lazily, inspect slices along a chosen dimension, and export them as separate TIFF files. Compatible with common bioimage formats, it simplifies handling complex datasets. (**b, c**) To fuse data acquired with our dual-view setup, we provide a napari plugin to assist users in manually defining a 3D rigid transformation either when it is trivial (e.g. a single axial translation) or to initialize an automatic registration tool close to the optimum. The first mode (left) allows the user to select salient landmarks and match them between the two views. An optimal rigid transformation is found automatically. The plugin's second mode (right) lets the user define each element of the transformation (translations and rotations) until a satisfactory match is observed in 3D or on 2D planes. (**d**) A complete preprocessing suite is provided as a napari plugin to transform raw 3D and 3D+ time datasets into datasets ready for analysis. The napari implementation allows for quick and visual exploration of parameters best suited to a given dataset. A recording feature can be toggled to save a complete user-defined pipeline into a *JSON* file, that can be used to process large datasets in batch or for sharing. (**e**) Interactive analysis of multiscale correlation heatmaps in napari is provided by allowing users to change the analysis length scale and to draw regions of interest directly on the correlation plot to see which cells contributed to the region's statistics on a 3D view of the data.

FoxA2. To further explore such transitions and localize specific cell populations, tools that allow interactive mapping of cell populations selected in the correlation plot to the actual gastruloid are needed. We have, therefore, integrated this functionality into our napari plugin (see *Figure 6*). We believe that our integrated imaging and analysis pipeline will allow researchers to study cell differentiation and cell state transitions at different stages of gastruloid development for the first time in full spatial context.

## Community-driven tools

We packaged our preprocessing and analysis pipeline into a *Python* library called *Tapenade* (for 'Thorough Analysis PipEliNe for Advanced DEep imaging'). The code is freely available on *GitHub* at github.

com/GuignardLab/tapenade/, *Vanaret et al., 2026*. The library comes with an extensive documentation and *Jupyter* notebooks accessible to non-specialist users. All datasets used to train our custom *StarDist* models to detect nuclei and mitotic events, along with the optimized weights for nuclei and mitotic events detection are currently available at https://doi.org/10.5281/zenodo.14748083. A non-specialist user can quickly segment a custom dataset by downloading the weights and loading them into one of the user-friendly implementations of *Stardist* like the *Stardist Fiji* plugin, the *napari* plugin *stardist-napari*, or the *ZeroCostDL4Mic* notebook (*Weigert et al., 2020*; *von Chamier et al., 2021*).

While working with large and dense 3D and 3D+ time gastruloid datasets, we found that being able to visualize and interact with the data dynamically greatly helped processing it. During the preprocessing stage, dynamical exploration and interaction led to faster tuning of the parameters by allowing direct visual feedback, and gave key biophysical insight during the analysis stage. We thus created four user-friendly napari plugins designed around facilitating such interactions (*Figure 6*):

a. *napari-file2folder* Efficient handling of large nD bioimaging datasets is a common challenge in bioimage analysis. To address this, we developed a napari plugin for lazy loading and efficient exploration of heavy bioimages using the zarr library. The plugin supports a wide range of standard bioimage file formats, including `lsm`, `tif`, `ome.tiff`, `zarr`, and `czi`. Upon lazy loading of an image, the user can immediately inspect its total shape across dimensions, even for datasets that cannot be fully loaded into memory. The plugin enables interactive selection of a single element along a specified dimension for focused exploration (e.g. viewing a single z-slice or a timepoint). Additionally, it offers a feature to save all elements along a selected dimension as separate `tif` files in a specified folder, which is particularly useful for downstream processing workflows with other plugins (*Figure 6a*). The plugin is freely available on the *napari* hub or at github.com/GuignardLab/napari-file2folder, *Vanaret, 2026b*.

b. *napari-manual-registration* When using our automatic registration tool to spatially register two views of the same organoid, we were sometimes faced with the issue that the tool would not converge to the true registration transformation. This happens when the initial position and orientation of the floating view are too far from their target values. We thus designed a napari plugin to quickly find a transformation that can be used to initialize our registration tool close to the optimal transformation. From two images loaded in napari representing two views of the same organoid, the plugin allows the user to either (i) annotate matching salient landmarks (e.g. bright dead cells or lumen-like structures) in both the reference and floating views, from which an optimal rigid transformation can be found automatically using singular value decomposition (*Arun et al., 1987*; *Figure 6b*), or (ii) manually define a rigid transformation by continually varying 3D rotations and translations while observing the results until a satisfying fit is found (*Figure 6c*). The plugin is freely available on the *napari* hub or at github.com/GuignardLab/napari-manual-registration, *Vanaret and Gros, 2026a*.

c. *napari-tapenade-processing* From a given set of raw images, segmented object instances, and object mask, the plugin allows the user to quickly run all preprocessing functions from our main pipeline with custom parameters while being able to see and interact with the result of each step (*Figure 6d*). For large datasets that are cumbersome to manipulate or cannot be loaded in napari, the plugin provides a macro recording feature: the users can experiment and design their own pipeline on a smaller subset of the dataset, then run it on the full dataset without having to load it in napari. The plugin is freely available on the *napari* hub or at github.com/GuignardLab/napari-tapenade-processing, *Vanaret and Gros, 2026b*.

d. *napari-spatial-correlation-plotter* This plugins allows the user to analyze the spatial correlations of two 3D fields loaded in napari (e.g. two fluorescent markers). The user can dynamically vary the analysis length scale, which corresponds to the standard deviation of the Gaussian kernel used for smoothing the 3D fields. If a layer of segmented nuclei instances is additionally specified, the histogram is constructed by binning values at the nuclei level (each point corresponds to an individual nucleus). Otherwise, individual voxel values are used. We took inspiration from the plugin *napari-clusters-plotter Zigutyte et al., 2023* in letting the user dynamically interact with the correlation heatmap by manually selecting a region in the plot. The corresponding cells (or voxels) that contributed to the region's statistics will be displayed in 3D on an independent napari layer for the user to interact with and gain biological insight (*Figure 6e*). The plugin

is freely available on the *napari* hub or at github.com/GuignardLab/napari-spatial-correlation-plotter, *Vanaret, 2026a*.

## Discussion

### Multiscale imaging and analysis pipeline

We have developed an integrated pipeline for *in toto* imaging of whole-mount gastruloids and automated image processing. A key innovation of our approach is its specific optimization for dense 3D tissues, such as gastruloids, which present similar imaging challenges and artifacts as tumors and spheroids, unlike many organoids that are shallow and less light-diffusive. Unlocking deep whole-mount imaging is crucial for researchers working with such dense tissues, as cryosectioning strategies cannot fully substitute for comprehensive three-dimensional (3D) *in toto* analysis (*van Ineveld et al., 2022*; *Costa et al., 2016*). Reconstructing 3D tissue microstructure from cryosectioned data is complex and challenging, involving slicing tissue into thin sections and imaging them individually. To achieve a 3D reconstruction, these 2D images must be meticulously aligned and stitched together, a process complicated by frequent issues, such as tissue deformation and section loss. In contrast, whole-mount imaging preserves spatial relationships and tissue integrity, making it the method of choice for studying cellular interactions and tissue architecture in their native context. Additionally, whole-mount imaging reduces sampling bias and enhances the detection of rare cell populations or phenotypic variations that might be missed in thin tissue sections. This limitation also apply to state-of-the-art spatial transcriptomic techniques, due to the fact that only few RNAs are typically captured per cell for each individual gene and many cells do not show any counts for a gene of interest expressed at low level (*Schott et al., 2024*).

### Application to other biological systems

In general, the pipeline is applicable to any tissue, but it is particularly useful for large and dense 3D samples, such as organoids, embryos, explants, spheroids, or tumors, that are typically composed of multiple cell layers and have a thickness greater than 50 *μm*. The processing and analysis pipeline are compatible with any type of 3D imaging data (e.g. confocal, 2 photon, light-sheet, live, or fixed). The applicability and limitations of the different modules of the pipeline are detailed *Table 1*.

Spectral unmixing to remove signal cross-talk of multiple fluorescent targets is typically more relevant in two-photon imaging due to the broader excitation spectra of fluorophores compared to single-photon imaging. In confocal or light-sheet microscopy, alternating excitation wavelengths often circumvents the need for unmixing. Spectral decomposition performs even better with true spectral detectors; however, these are usually not non-descanned detectors, which are more appropriate for deep tissue imaging. Our approach demonstrates that simultaneous cross-talk-free four-color two-photon imaging can be achieved in dense 3D specimen with four non-descanned detectors

**Table 1.** Purpose, requirements, applicability, and limitations of each step in the processing and analysis pipeline.

| | Spectral filtering | Custom 3D nuclei segmentation | Normalization | Multi-scale analysis |
|---|---|---|---|---|
| Purpose | Generate cross-talk free four-color images of markers imaged simultaneously | Quantifying cell-scale gene expression and nuclei morphological properties | Correct for intensity gradients due to optical artifacts in 3D | Analyze gene expression patterns, cellular events, and morphologies in 3D across multiple spatial scales |
| Requirements | Calibration of emission spectra as a function of imaging depth | Nuclei must be star-convex and ~15 px per nucleus after resampling | At least one ubiquitously expressed channel; calibration for wavelength-dependent correction | If domain-based signal (e.g. gene expression signals): domain expression mask. If object-based signal (e.g. morphological features) instance masks. |
| Applicability | Mostly useful in two-photon imaging because of broad excitation spectra | Tested on tissues from multiple species and modalities (*Figure 3—figure supplement 2d*) | Applicable to any multichannel data across tissues and modalities | Any 3D data with gene expression or morphological information |
| Limitations | / | Small (<15 px/nucleus) and anisotropic resolution may impair segmentation | / | Hollow structures may bias results if no proper masking is available |

and co-excitation by just two laser lines. Depending on the dispersion in optically dense samples, depth-dependent apparent emission spectra need to be considered. Spectral filtering has already been applied in other systems (e.g. *Rakhymzhan et al., 2017* and *Dunsing et al., 2021*), but is here extended to account for imaging depth-dependent apparent emission spectra of the different fluorophores. In our pipeline, we provide a code to run spectral filtering on multichannel images, integrated in Python. In order to apply the spectral filtering algorithm utilized here, spectral patterns of each fluorophore need to be calibrated as a function of imaging depth, which depends on the specific emission windows and detector settings of the microscope.

Nuclei segmentation using our trained StarDist3D model is applicable to any system under two conditions: (1) the nuclei exhibit a star-convex shape, as required by the StarDist architecture, and (2) the image resolution is sufficient in XYZ to allow resampling. The exact sampling required is object- and system-dependent, but the goal is to achieve nearly isotropic objects with diameters of approximately 15 pixels while maintaining image quality. In practice, images containing objects that are natively close to or larger than 15 pixels in diameter should segment well after resampling. Conversely, images with objects that are significantly smaller along one or more dimensions will require careful inspection of the segmentation results. To evaluate our 3D nuclei segmentation model, we tested it on diverse systems, including gastruloids stained with the nuclear marker Draq5 from *Moos et al., 2024*; breast cancer spheroids; primary ductal adenocarcinoma organoids; human colon organoids and HCT116 monolayers from *Ong et al., 2025*; and zebrafish tissues imaged by confocal microscopy from *Li et al., 2023*. These datasets were acquired using either light-sheet or confocal microscopy, with varying imaging parameters (e.g. objective lens, pixel size, staining method). Given the difficulty to access ground truth annotations of 3D organoid data and the multiplicity of models in the literature, we only show qualitatively the versatility of our segmentation model (*Figure 3—figure supplement 2*). The results show satisfactory performance on these different datasets and imaging conditions.

Normalization is broadly applicable to multicolor data when at least one channel is expected to be ubiquitously expressed within its domain. Wavelength-dependent correction requires experimental calibration using either an ubiquitous signal at each wavelength. Importantly, this calibration only needs to be performed once for a given set of experimental conditions (e.g. fluorophores, tissue type, mounting medium). To our knowledge, the calibration procedures for spectral-filtering and our image-normalization approach have not been performed previously in 3D samples, which is why validation on published datasets was not possible. Nevertheless, they are described in detail in the Methods section, and the code used from the calibration measurements to the corrected images is available open-source at the Zenodo link in the manuscript.

Multi-scale analysis of gene expression and morphometrics is applicable to any 3D multicolor image. This includes both the 3D visualization tools (Napari plugins) and the various analytical plots (e.g., correlation plots, radial analysis). Multi-scale analysis can be performed even with imperfect segmentation, as long as segmentation errors tend to cancel out when averaged locally at the relevant spatial scale. However, systematic errors, such as segmentation uncertainty along the Z-axis due to strong anisotropy may accumulate and introduce bias in downstream analyses. Caution is advised when analyzing hollow structures (e.g. lumen or curved epithelial monolayers with large cavities), as the pipeline was developed primarily for 3D bulk tissues, and appropriate masking of cavities would be needed.

In this work, we have not used refined masks of the gastruloids that would account for and exclude hollow structures. In *Figure 4a and c*, we noted that this lead to noticeable patterns in the packing-related coarse-grained fields and in the local alignment of the true strain tensor major axis. To improve our current implementation, our masking approach could be augmented by a second method used to specifically detect these structures. Pixel classification based on Random-Forests could be used to get accurate segmentation using only a few manually annotated lines. These approaches are now widely accessible to non-specialist users in softwares like *Fiji* (with *Labkit Arzt et al., 2002*), *Ilastik Berg et al., 2019*, or *napari* (with *napari-APOC Haase et al., 2023*).

## A Python integrated pipeline for non-specialists, with napari-enabled 3D exploration

We built our pipeline in Python to leverage its rich ecosystem of scientific computing libraries while keeping the code as accessible and user-friendly as possible. We provide detailed installation

instructions and Jupyter notebooks with step-by-step instructions designed to be used by non-specialists, with a focus on ease of use and reproducibility. To tackle the challenges that arise from manipulating deep and dense 3D data, we provide napari plugins for manual registration, preprocessing, and analysis. napari provides a user-friendly graphical interface which enables real-time visualization and exploration of 3D data during each step of the preprocessing and analysis. This was essential for the development of the pipeline, as the direct visual feedback allowed us to quickly iterate and optimize the processing steps, while gaining insights into the 3D organization of gastruloids. We believe our pipeline offers a distinctive and effective integration of 3D data exploration and processing, providing a valuable tool for the organoid and tumoroids research community.

We did not explore continuously adjusting laser power based on excitation depth to enhance signals in deeper tissue layers, which could further extend the depth limits of our imaging method. In principle, incorporating such adaptation should not affect our pipeline, as long as the laser used for exciting the nuclei channel (which normalizes all other signals) is the same or maintains the same power-to-depth relationship as the lasers exciting the other fluorophores to be normalized. Here, we developed a wavelength-dependent normalization scheme based on a simplified columnar model. Future work could focus on creating a complete model that accounts for photon paths in a complex geometry, for example, in elongated gastruloids, and integrate the spectral filtering approach to the wavelength-dependent normalization scheme.

In this work, we assume that the nuclear signal is the most reliable for normalizing other signals due to its extensive coverage of the tissue (over 60% in gastruloids). Instead of normalizing signals at the level of individual nuclei, we developed a correction scheme that identifies the optimal scale of characteristic gradients in nuclear intensities to generate a continuous field for re-normalization. This approach has the added advantage of being applicable to markers with different subcellular localizations (e.g. nuclei, cell membrane, intracellular organelles).

A necessary component of our quantitative imaging pipeline was the use of spectral unmixing to address spectral cross-talk, an inherent challenge in multi-color acquisitions, particularly with two-photon excitation. This allowed us to image four different fluorescent targets simultaneously and was required to capture and correlate multiple genetic markers on the single-cell level. This can be further expanded in the future by using state-of-the-art hyperspectral detection, which allows capturing up to 32 channels simultaneously and thus decompose even more spectrally overlapping fluorescent targets (*Cutrale et al., 2017*). Two-photon excitation is advantageous in this context, allowing for the excitation of multiple targets with a single laser line. In our work, three of the four fluorophore species (Hoechst, AF488, and AF568) were excited using a 920 nm laser line. Additionally, iterative rounds of immunostaining and washing can be applied to maximize the information content (*Zheng et al., 2023*; *Bolognesi et al., 2017*). In this context, the interactive registration tools that we have implemented here will be key ingredients to correct for sample movements between subsequent staining steps.

## Exploring gastruloid development in 3D

In the context of gastruloid development, which serves as the central biological theme of this paper, it is crucial to understand how spatial heterogeneities in gene expression arise at both the cell microenvironment scale and the larger gastruloid scale. During early gastruloid symmetry breaking and elongation, radial and later antero-posterior gradients of the gene T-Bra, a nascent mesoderm marker, emerge. These radial gradients can be captured with our coarse-grained analysis (see *Figure 5c*). However, while both the endoderm marker FoxA2 and T-Bra exhibit radial gradients at the coarse-grained scale, their anti-correlated expression is only evident at the single-cell level (see *Figure 5c–f*). This example highlights the importance of a multiscale analysis pipeline that allows simultaneous visualization of tissue-scale properties and local cellular events. Our methodology utilizes segmentation of the Hoechst-stained nuclei channel to (1) identify individual cells for cell-scale analysis and (2) normalize other channels using the nuclei signal. By developing a diverse training set for nuclei segmentation, we achieved precise determination of nuclei shapes. This precision enables systematic studies of the relationship between nuclear morphology and gene expression patterns, and can be extended to cell morphology using membrane stains, such as ZO-1 and cell shape segmentation methods like CellPose (*Stringer et al., 2021*; *Pachitariu et al., 2025*). Note that the latest version of the latter method, CellPoseSAM (*Pachitariu et al., 2025*), can give satisfactory results for nuclei

segmentation, and we showed qualitatively that it benefits from the Tapenade preprocessing workflow for segmenting deep tissues (*Figure 3—figure supplement 3a*) and that using it as a replacement for our trained StarDist3D model lead to similar results for the nuclear morphology study (*Figure 3— figure supplement 3b–d*).

Nuclei deformation patterns show large-scale alignment at the gastruloid level (see *Figure 4c–d*) and localized alignment near lumens and cavities, providing potential insights into the structuring of epithelial and mesenchymal tissues during gastruloid development. In older gastruloids with cavity-containing structures, such as somitic structures (*Veenvliet et al., 2020*), automated cavity detection will be crucial for accurately quantifying coarse-grained fields of cell or division densities.

While this work focuses on gastruloid datasets, we believe that the pipeline effectively extends to organoid, tumoroid, and other datasets that encounter challenges related to whole-mount deep imaging. We designed the library in the prospect of long-term maintenance, and it will receive continuous improvements in the future.

# Materials and methods
## Sample preparation
### Gastruloid culture

Gastruloids were generated using the protocol described previously in *Hashmi et al., 2022*, from four different cell lines. Briefly, cells were seeded and aggregated for 48 hr in low-adherence 96-well plates (Costar ref: 7007) and subsequently pulsed with the Wnt agonist Chiron, which was washed out after 24 hr, i.e., at 72 hr of aggregate culture. Time points indicated throughout the paper refer to the time after seeding. The two cell lines T-Bra-GFP/NE-mKate2 and E-cad-GFP/Oct4-mCherry were used for gene expression analyses. The E14Tg2a.4 line (ATCC) was used for spectral calibration and benchmarking, as it is devoid of any fluorescent construct. The H2B-GFP (a generous gift from Kat Hadjantonakis) was used for live imaging experiments and morphometric analyses. In brief, between 50 and 400 cells were seeded in low adhesive 96 well plates in a neural differentiation medium. In order to promote endoderm formation and to reduce symmetry breaking variability compared to the standard gastruloids protocol (*van den Brink et al., 2014*), we enriched the medium with Fibroblast Growth Factor and Activin as in *Hashmi et al., 2022*. All cell lines were tested to be free from mycoplasma contamination using qPCR.

### Immunostaining

Samples were immunostained using the same protocol as in *Hashmi et al., 2022* apart from the fact that samples were incubated in a 1 mM glycine solution, and that primary antibodies (see table below) were incubated during three consecutive days. Secondary antibodies AF488, A568, AF647 (Thermo Fisher) and Hoechst 33342 (Thermo Fisher) were chosen to be compatible with four-color two-photon imaging. They were diluted 500 times for incubation, except for Hoechst which was diluted 10,000 times in order to minimize the crosstalk of the Hoechst signal into other channels (*Table 2*).

**Table 2.** Antibodies.

| Antibody | Species | Reference | Provider | Dilution |
|---|---|---|---|---|
| Brachyury | Goat | AF2085 | RD system | 1:40 |
| E-cadherin | Rat | M108 | Takara | 1:200 |
| FoxA2 | Goat | sc-6554 | Santa-Cruz | 1:50 |
| Snail | Goat | AF3639 | RD system | 1:50 |
| OB-cadherin | Rabbit | 71–7600 | Invitrogen | 1:500 |
| pH3 | Rat | ab10543 | abcam | 1:250 |
| Oct4 | Mouse | MAB4419 | Merck Millipore | 1:100 |
| Sox2 | Mouse | sc-365823 | SantaCruz Biotech. | 1:100 |
| anti-GFP | Chicken | GFP-1020 | Aves | 1:100 |

## Mounting

Samples were mounted between two glass coverslips of standard thickness (0.31 to 0.17 mm) using 250 $\mu m$ or 500 $\mu m$ spacers (SUNJin Lab, Taiwan, R.O.C.) in 20 $\mu l$ or 40 $\mu l$ mounting medium. For the clearing benchmark, gastruloids were incubated overnight and then mounted in either PBS, 80% glycerol solution in PBS (v/v) (*Ahmad et al., 2021*), 20% optiprep solution in PBS (v/v) (*Boothe et al., 2017*), or antifade mounting medium (ProLong Gold Antifade Mountant, Thermo Fisher Scientific). For all other experiments with fixed samples, gastruloids were mounted in 80% glycerol solution in PBS (v/v). For live imaging, gastruloids were transferred from 96-well plates to either MatTek dishes (MatTek corporation, ref: P35G-1.5–14 C) or to micro-well plates (500 $\mu m$ wells, SUNBIOSCIENCE ref: Gri3D) in culture medium and imaged in a chamber maintained at 37°C, 5% $Co_2$ with a humidifier.

## Two-photon imaging

Four-colors two-photon imaging of immunostained samples was performed on an upright Nikon A1 R MP microscope using an Apo LWD 40x/1.1 WI $\lambda$ S DIC N2 objective. Fluorescence was simultaneously excited at 920 nm (to excite simultaneously Hoechst, AF488, and AF568) with a tunable femtoseconde laser (Chameleon Discovery NX from Coherent) and at 1040 nm (to excite simultaneously AF568 and AF647) with the same laser secondary output at fixed wavelength. Fluorescence was detected on four non-descanned GaAsP detectors with the following emission filters: 450/70 (Ch1), 530/55 (Ch2), 607/70 (Ch3), 700/50 (Ch4). To spectrally separate the channels, three dichroic mirrors (488 LP, 562 LP, 650 LP) were used. Multiposition imaging was used to automatically acquire image stacks on multiple gastruloids mounted in the same sample slide.

Live imaging of gastruloids was performed on a Zeiss 510 NLO (Inverse - LSM) with a femtosecond laser (Laser Mai Tai DeepSee HP) with a 40x/1.2 C Apochromat objective. The Histone-GFP signal was excited at 920 nm and detected with a non-descanned GaAsP detector with a 560 LP. The images were acquired with the full field-of-view (318 $\mu m$, pixel size 0.62 $\mu m$), and a z-spacing of 1 $\mu m$. Depth in z varies from 150 to 350 $\mu m$, depending on the acquisitions and the sample size, but always past the midplane of the gastruloid.

## Confocal imaging

Confocal imaging was performed on a Zeiss LSM880 system (Carl Zeiss) using a Plan-Apochromat 40x/1.2 NA water immersion objective. Fluorescence was excited with the 488 nm line of an Argon laser. To split excitation and emission light, a 488 dichroic mirror was used. Fluorescence was detected between 491 and 700 nm on a 32-channel GaAsP array detector, with a pinhole set to one airy unit.

## Image quality estimation

To quantitatively assess clearing efficiency and compare the information content of images at depth, we used the Fourier ring correlation quality estimate (FRC-QE) (*Preusser et al., 2021*). We selected a column of 400×400 pixels for the clearing benchmark and 100×100 pixels for all other analyses in the center of aggregates and computed FRC-QE using the available Fiji plug-in.

## Preprocessing: Spectral unmixing

To remove spectral cross-talk, four-channel images were spectrally unmixed into $n$ images corresponding to the contribution from each of the $n$ fluorophore species present in the sample.

### Spectral calibration

To this aim, reference spectra $p_i^k$ of the utilized fluorophores were determined from calibration images acquired on aggregates stained with only one fluorophore species at the same excitation and detection settings. To account for wavelength-dependent signal decay with imaging depths, spectral patterns were computed as a function of imaging depth. It is important to note that spectral patterns depend on the gain settings of each detector. However, once calibrated for a certain set of gain values, reference patterns for different gains can be easily interpolated by simply measuring the count ratios for both sets of gains on a sample providing sufficient signal (i.e. well above background counts) on each detector for all gains.

## Spectral decomposition

Four-channel images $I^k(x, y, z)$ were decomposed using a previously published unmixing algorithm originally developed in the context of fluorescence lifetime correlation spectroscopy (*Ghosh et al., 2018*), image correlation (*Schrimpf et al., 2018*), and fluorescence correlation spectroscopy (*Benda et al., 2014*; *Dunsing et al., 2021*). First, spectral filter functions $f_i^k$ were calculated using the calibrated reference spectra $p_i^k(z)$:

$$f_i^k(z) = \left( \left[ \hat{P}^T D \hat{P} \right]^{-1} \hat{P} D \right)_{ik} \tag{1}$$

Here, $P$ is a matrix with elements $P_{ki} = p_i^k(z)$ and $D$ a diagonal matrix computed from the average image intensity in each channel $k$, $D = diag \left[ 1/\langle I^k(x, y, z) \rangle_{x,y} \right]$.

The spectral filter functions represent intensity weights per spectral channel. To obtain the filtered images $I_i(x, y, z)$ for each fluorophore species $i$, the channel images are multiplied with the filter functions and summed over all channels ($m = 4$ here):

$$I_i(x, y, z) = \sum_{k=1}^{m} f_i^k(z) I^k(x, y, z). \tag{2}$$

It should be noted that unmixing successfully works as long as signal levels are well above background counts and all images, acquired for spectral calibration and in actual imaging experiments, are devoid of detector saturation. This way, even closely overlapping fluorophore species can be discriminated with only few detector channels, at the expense of increased image noise (*Schrimpf et al., 2018*). In this context, pure detector channels, i.e., channels that predominantly detect one of the fluorophore species, are beneficial. This is the case in the example of GFP and AF488 presented in *Figure 2—figure supplement 1e and f*, in which the first channel detects mostly GFP fluorescence. In addition, samples in which signal from different fluorophore species is detected in different pixels (e.g. pixels corresponding to nuclei and membranes) are generally easier to unmix and suffer less from noise increase than samples were different fluorophore species contribute to the signal in the same pixels.

## Preprocessing: Dual-view registration and fusion

Multiple organoids were acquired from the same slide, first from one side for all of them, then from the other side. Therefore, the first step of the registration and fusion algorithm was to pair the two sides of each organoid together. This pairing was done by extracting the sample positions stored in the metadata of the acquisitions and finding the pairing that minimizes the sum of the distances between paired samples using a linear assignment algorithm.

Registration of these dual-view image stacks was achieved using an existing intensity-based block-matching algorithm (*Ourselin et al., 2000*), adapted for 3D microscopy images and previously used in *McDole et al., 2018*; *Guignard et al., 2020*. From the two distinct images, one is arbitrarily chosen as the reference image and the other, named the floating image, is registered onto the reference one. Starting from an initial transformation, the blockmatching algorithm compares, on multiple hierarchical levels, each block in the floating image with the corresponding neighboring blocks in the reference image, and finds the rigid 3D transformation (specified by rotation around axes X, Y, Z, and translation along axes X, Y, Z) from these sets of blocks. Then, we apply the transformation to the floating image, so that the two views are in the same referential, and fuse them into one image stack.

Because of the computational cost of comparing blocks in large 3D images, the size of the neighborhood that the registration algorithm explores is restricted. Therefore, the magnitude of the transformation between the two image stacks has to be relatively small (typically less than 30° rotations and 100 units in translation). As a consequence, the user has to provide an initial transformation to the algorithm, which approximates the actual transformation. In the case of dual-view imaging with opposite views, the initial transformation would correspond to 180° rotation around the X axis and some additional translation. To explore the initial transformation, we developed the napari plugin *napari-manual-registration* detailed above and illustrated in *Figure 6*. This tools allows to interactively rotate and translate the floating with respect to the reference image stack. The tool also allows users to specify manual landmarks in both image stacks, e.g. lumen, dead or dividing cells, from which an

initial transformation can be automatically computed. While the blockmatching algorithm generally allows for any type of transformation (rigid, affine, non-linear, …), we restrained the search space to rigid transformations. Moreover, since the stacks comprise multiple channels, we used a reference one that was expressed ubiquitously, here the nuclei staining, on which the transformation was computed. The other channels were registered using the same transformation matrix as the reference channel. Because the SNR of each stack decreased roughly with the distance to the objective, we weighted the contribution in the fusion of the two opposing sides of the sample. The weights $f_1$ and $f_2$, respectively, for the reference view and floating view were computed as a sigmoid:

$$f_1(z) = \frac{1}{1 + e^{p(z-z_0)}},$$  (3)

$$f_2(z) = 1 - f_1(z).$$  (4)

with $z$ being the depth in the stack in voxels, $z_0$ the middle (i.e. inflection point) of the sigmoid, where $f_1(z_0) = 0.5$, and $p$ the slope of the sigmoid. The decay length $l$ of the sigmoid and the effective fusion width $\delta$ are linked by $l = 4/p$ and $\delta = 2l$.

As illustrated in *Figure 2e*, the two sides were imaged up to a certain depth depending on the sample size and the image quality, past the mid-plane to ensure some overlap with the other view. Because we acquired the same depth and number of z-planes for both views, the sigmoid is centered in the middle of the overlapping region, which is why we considered $z_0 = 0.5$. The overlapping region to fuse had a typical width value of $w = 70$ to $150\ \mu m$ (depending on the image depth) around the midplane. The coordinate along the overlap region was mapped to the range [0,1] to allow setting the parameters of the sigmoid fusion function independently from the size of the overlap or the size of the gastruloid, as illustrated on *Figure 2—figure supplement 2*. The slope of the sigmoid was set to $p = 15$, which leads to $l = 0.27$ and $\delta = 0.53$. Intuitively, this means that 53% of the overlapping region was used to mix both signals with non-negligible mixing coefficients to create the fused part.

## Nuclei segmentation

To segment nuclei, we trained a custom Stardist3D (*Weigert et al., 2020*) model on three annotated datasets acquired with two-photon microscopy in different acquisition modalities: one fixed dataset, two live datasets (one with mosaic fluorescence labeling), see *Table 3* and *Figure 3—figure supplement 1a*.

### Global contrast enhancement

Before training and inference with our Stardist3D model, we applied percentile-based histogram clipping and normalization to the input images. For a given image voxel at position $\vec{x} = (z, y, x)$, its intensity $I(\vec{x})$ is first transformed as follows:

$$I'(\vec{x}) = \frac{I(\vec{x}) - P_{min}^{image}(\vec{x})}{P_{max}^{image}(\vec{x}) - P_{min}^{image}(\vec{x})},$$  (5)

where $P_{min}^{image}(\vec{x})$ and $P_{max}^{image}(\vec{x})$ are low and high percentile intensity values in the image.

In practice, we choose $P_{min}^{image}$ and $P_{max}^{image}$ to be the first and 99th percentile of the intensity values in the image. As shown *Figure 3—figure supplement 1d*, applying global contrast enhancement was not sufficient to restore sufficient intensity in the deeper regions of some samples, which lead to poor segmentation performances. We thus extended our contrast enhancement method to apply in local sub-regions of the images.

**Table 3.** Description of the datasets used for training our custom StarDist3D model to detect nuclei. All datasets were acquired with a two-photon microscope.

| Density | Dimension | Num. nuclei | Pixel size (µm/pix) |
|---------|-----------|-------------|---------------------|
| Dense | 3D | 1018 | Z: 1, Y/X: 0.25 |
| Dense | 3D+time | 100 | Z: 1, Y/X: 0.62 |
| Sparse | 3D+time | 3296 | Z: 1, Y/X: 0.62 |

## Local contrast enhancement

Because the image contrast is not homogeneous within and across samples, we homogenized intensity profiles across the training datasets by applying local percentile-based histogram clipping and normalization.

For a given image voxel at position $\vec{x} = (z, y, x)$, its intensity $I(\vec{x})$ is first transformed as follows:

$$I'(\vec{x}) = \frac{I(\vec{x}) - P_{min}^{window}(\vec{x})}{P_{max}^{window}(\vec{x}) - P_{min}^{window}(\vec{x})}, \tag{6}$$

where $P_{min}^{window}(\vec{x})$ and $P_{max}^{window}(\vec{x})$ as before, the first and 99th percentile of the intensity values within the window, and $w$ close to the average size of the nuclei in the dataset. The transformed image $I'$ is finally clipped to the range [0,1] to obtain the locally enhanced image.

The training datasets were rescaled to an isotropic voxel scale of 0.62 $\mu m$/pix in all directions, from which crops of 64×64×64 voxels were extracted and augmented during training by 3D rotations, flipping, intensity shifts, and additive Gaussian noise.

Applying scaling transformations to crops during training is often regarded as the best way to achieve robustness in the case of a wide object size distribution. This is typically the case when several raw datasets acquired in different experimental conditions are combined, particularly due to different pixel sizes across datasets. However, we found in our case that the size distribution related to a single dataset is narrow enough that robustness is better achieved by first rescaling all training datasets to a common isotropic nuclei size. (*Stringer et al., 2021*; *Vanaret et al., 2023*).

Before inference, a new dataset simply needs to be rescaled to reach that common nuclei size.

## Mitotic cells segmentation

Another StarDist model was trained to detect the divisions stained by ph3. To this aim, we iteratively trained StarDist3D models with a very small batch of annotated data, using the model trained on all nuclei, and corrected the prediction using napari to train again. At the end, 31 images of whole-mount gastruloids and 5300 divisions were used to train a custom Stardist3D model, using the same parameters as for the nuclei detection model in terms of preprocessing, crops size, and data augmentation.

Whereas the model trained on nuclei was not very successful on ph3-stained nuclei, because of their particular shapes, the final model after multiple iterations achieved better results, around F1=82%. Some over-segmentations around large and deformed nuclei were still present, and using 3D Gaussian blur did not achieved a better score, because they led to misdetections where neighboring nuclei were in mitosis. Therefore, after prediction, post-processing was used to filter out small volumes with thresholding. Events detected from the background or dead cells (using a mask of the sample computed from the nuclei channel) and minimal manual curation was done, using napari.

## Masked Gaussian convolution to probe spatial fields at different scales

To probe volumetric signals at different length scales $\sigma$ ranging from the average nuclei size to the width of the organoid, we convolve signals with a Gaussian kernel of size σ. We propose two ways to compute convolutions depending on whether the signal is dense (e.g. an image representing a signal intensity) or sparse (e.g. nuclei volumes defined only locally on nuclei centroids).

### Dense data

We apply convolution with a normalized Gaussian kernel to dense signals. For a given voxel at position $\vec{x} = (z, y, x)$, its value $I(\vec{x})$, whether scalar or vector, is transformed according to:

$$I^{\sigma}(\vec{x}) = \sum_{\vec{x}' \in [-a,a]^3} k_{\sigma}(\vec{x}') I(\vec{x} - \vec{x}'), \tag{7}$$

with $k_{\sigma}(\vec{x}') = (2\pi\sigma^2)^{-3/2} \exp\left(-\left\|\vec{x} - \vec{x}'\right\|^2 / 2\sigma^2\right)$, and $a$ the convolution window size. In our implementation, we choose $a = 3\sigma$.

When a binary mask (e.g. defining the inside of the organoid) is available, we apply a masked convolution with a correcting normalization factor. Let $M$ be a binary masking function, e.g., $M(\vec{x}) = 1$ for voxels at positions $\vec{x}$ inside the organoid, and 0 everywhere else, we compute

$$I^{\sigma,M}(\vec{x}) = \frac{\sum_{\vec{x}' \in [-a,a]^3} k_\sigma(\vec{x}')I(\vec{x}-\vec{x}')M(\vec{x}-\vec{x}')}{\sum_{\vec{x}' \in [-a,a]^3} k_\sigma(\vec{x}')M(\vec{x}-\vec{x}')}. \tag{8}$$

This additional normalization factor prevents the artifactual decay of the convolved signal close to the mask boundaries by 'excluding' voxels outside the mask from the convolution. Note that the masked formulation allows the user to compute values of a quantity outside its domain of expression. For instance, if both a mask $M_{inst}$ of nuclear instances and a mask $M_{in/out}$ that marks the inside and outside of the sample are provided, a quantity like the nuclear Hoechst expression can be computed in the full region defined by $M_{in/out}$, even though it only appears in the nuclei.

## Sparse data

For sparse data, i.e., quantities defined locally at $n$ specific positions $\mathbf{X} = \{\vec{x}_1, \vec{x}_2, \ldots, \vec{x}_n\}$, we compute the convolution and masked convolution in a similar manner, but with a mask $M$ that is non-zero only at the positions at which the sparse signal is defined (e.g. nuclei centroids). This formulation allows the user to choose between studying the sparse signal (i) at its original positions, i.e., by computing the above convolution at positions $\vec{x} \in \mathbf{X}$, (ii) at resampled positions, e.g., on a uniform grid for convenience, or (iii) as a dense continuous field across the whole domain of the gastruloid.

We provide a second implementation of the masked convolution specifically for sparse data, which is more convenient when the smoothed sparse signals should also be smooth (e.g. when it should be defined at its initial positions or on a regular grid). This implementation uses the Kd-Tree data structure implemented in *Scipy* to efficiently find neighbor relationships from the positions at which the sparse signal is defined, which is used to optimize the computation of the convolution (*Equation 8*).

## Preprocessing: Intensity normalization

Quantifying gene expression in 3D tissue samples from imaging data is hindered by various sources of noise and optical artifacts that induce large scale intensity gradients (e.g. scattering, aberrations). The main sources of artifacts are (local) variations in optical paths and cell density, that usually manifest as decreased intensities towards deep tissue regions. Artifactual gradients can also appear during the registration of two non-collinear views, which further complexifies the expected distribution of artifacts and resulting intensity gradients.

We model the intensity signal $I(\vec{x})$ from a given channel as

$$I(\vec{x}) = M_{inst}(\vec{x})S(\vec{x})A(\vec{x}) + \varepsilon(\vec{x}), \tag{9}$$

where $M_{inst}(\vec{x})$ is a binary mask of the signal instances (e.g. nuclei or membranes), i.e., $M_{inst}(\vec{x}) = 1$ if $\vec{x}$ is inside an instance and 0 otherwise, $S(\vec{x})$ is the gene expression field, which provides a continuous representation of the cell-scale variations of the biological signal, $A(\vec{x})$ is the field representing the artifactual intensity variations, and $\varepsilon(\vec{x})$ is an additive uncorrelated noise term which models both variations in the signal at scales below the cell-scale and acquisition noise. This decomposition is illustrated on *Figure 5—figure supplement 2*.

We hypothesize that the Hoechst signal $S_h(\vec{x})$ is globally homogeneous throughout the volume, i.e $S_h(\vec{x}) = S_{h,0}$, and that the field of optical artifacts $A$ has large scale variations (typically superior to a few cell diameters). In particular, direct observations of homogeneous signals like Hoechst with different emission wavelengths, as displayed of *Figure 5—figure supplement 1a–b*, indicate that optical artifacts are mostly dominated by exponential decay with depth with a wavelength-dependent decay length. A second order contribution comes from scattering effects related to heterogeneities in photon's optical paths, cellular density, or to geometrical considerations, e.g., proximity to the sample's boundaries. For every wavelength $\lambda$ and every signal, we thus model $A$ to first order as

$$A(\vec{x}, \lambda) = e^{-z/d} + a(\vec{x}, \lambda), \tag{10}$$

with $d$ the wavelength-dependent decay length. The second term $a$ on the right hand side corresponds to second order corrections to the exponential decay model. Note that, in accordance with Beer-Lambert law, the value $d(\lambda)$ also likely depends on experimental parameters $\mathcal{E}$, like the nature of the mounting medium or the settings of the experiments, in a multiplicative fashion, i.e.,

$$d(\lambda, \mathcal{E}) = f(\mathcal{E})d'(\lambda), \tag{11}$$

so that in similar imaging scenarios, ratios of decay lengths at two wavelengths

$$\frac{d(\lambda_2, \mathcal{E})}{d(\lambda_1, \mathcal{E})} = \frac{d'(\lambda_2)}{d'(\lambda_1)} = r(\lambda_2, \lambda_1) \tag{12}$$

only depend on $\lambda_1$ and $\lambda_2$.

To estimate the decay lengths across the range of wavelengths in use, we measured the intensity decay in samples with ubiquitous nuclei stainings in four emission wavelengths: Hoechst for the blue channel ($\lambda = 405\,\mathrm{nm}$), H2B-GFP amplified for the green channel ($\lambda = 488\,\mathrm{nm}$), SPY-DNA555 or the endogenous H2B-Tomato for the red channel ($\lambda = 555\,\mathrm{nm}$) and DRAQ5 for the far-red channel ($\lambda = 647\mathrm{nm}$). Stainings for different wavelengths were done on separated samples to avoid signal crosstalk between the channels. All samples were imaged with the same excitation and detection settings. For each sample, we extracted the intensity profile in a central column along the z-axis to minimize boundary artifacts, and fitted *Equation 10* to recover the decay lengths $d(\lambda)$ (*Figure 5— figure supplement 1a–c*). We observe a monotonous relationship between median values of $d$ across different samples and $\lambda$ . To confirm our measurements, we also computed decay lengths via a second method: notice that the model of *Equation 10* leads to the following identity for two ubiquitous signals $I_1$ and $I_2$ at wavelengths $\lambda_1$ and $\lambda_2$ (neglecting the noise term $\varepsilon$ in *Equation 9* and the second order term from *Equation 10*)

$$\log(I_1) = r(\lambda_2, \lambda_1)\log(I_2) + cst. \tag{13}$$

We imaged new samples with both Hoechst ($\lambda = 405\,\mathrm{nm}$) and the Far-red nuclei stain ($\lambda = 647\,\mathrm{nm}$). There was no significant spectral overlap between the two signals. By fitting *Equation 13*, we obtained another estimate of the ratio $r(\lambda = 405\,\mathrm{nm}, \lambda = 647\,\mathrm{nm}) \approx 0.46$, consistent with the value of 0.48 obtained with the previous method (*Figure 5—figure supplement 1e*). In *Figure 5—figure supplement 1d*, we plot the decay length ratios with the decay length of the Hoechst channel $d_{blue}$. In our model, these ratios are independent of imaging conditions, and the monotonicity and smoothness of the trend suggests that the ratios $d_{blue}/d(\lambda)$ in the range $\lambda \in [405\,\mathrm{nm}, 647\,\mathrm{nm}]$ could be estimated by interpolating our results.

In our experiments, it was not possible to image ubiquitous nuclei signals at the same wavelengths as relevant biological signals, while Hoechst staining was often easily available. We thus developed a correction scheme based on Hoechst staining to separate the contribution of artifactual gradients from true gradients in gene expression. The rationale of the scheme stems from the fact that *Equation 13* also allows us to relate two ubiquitous signals up to a multiplicative constant:

$$I_1 \propto I_2^{r(\lambda_2, \lambda_1)}. \tag{14}$$

The Hoechst signal could, therefore, be used to predict the intensity profile from another 'virtual' ubiquitous signal imaged at some wavelength $\lambda \neq \lambda_{blue}$ that coincides with the wavelength of the biological signal of interest.

First, we compute the Gaussian average of the reference Hoechst signal $I_h$ at a scale $\sigma$ using *Equation 8*, in which we keep only voxel intensities that belong to the nuclei using a mask $M_{nuc}$:

$$I_h^{\sigma, M_{nuc}}(\vec{x}) \approx S_{h,0}A(\vec{x}). \tag{15}$$

Here, $\sigma$ is chosen large enough to average out the additive noise $\varepsilon$ but small enough that the structure of the field of optical artifacts $A$ is preserved. The practical choice of $\sigma$ is discussed below.

We then normalize other biologically relevant intensity signals $I_b$ by the Gaussian-averaged nuclei field $\left(I_h^{\sigma, M_{nuc}}(\vec{x})\right)^{r(\lambda_{blue}, \lambda_b)}$ to get

$$
\begin{aligned}
\hat{I}_b(\vec{x}) &= \frac{I_b(\vec{x})}{\left(I_h^{\sigma, M_{nuc}}(\vec{x})\right)^{r(\lambda_{blue}, \lambda_b)}} \\
&= M_{inst}(\vec{x})\frac{S(\vec{x})}{S_{h,0}^{r(\lambda_{blue}, \lambda_b)}} + \varepsilon'(\vec{x}),
\end{aligned}
\tag{16}
$$

with $\varepsilon'$ the normalized version of the additive noise $\varepsilon$. This step effectively removes the field of optical artifacts from the signal. The choice of $\sigma$ used for the Gaussian averaging of the ubiquitous signal depends on the length scale at which the field of artifacts $A$ varies. Since this is not known in practice, we choose in our implementation the value $\sigma$ that leads to the most homogeneous normalized ubiquitous signal across all z-planes. Formally,

$$\begin{cases} \sigma = \mathrm{argmin}_\sigma \, \mathrm{MAD}\left(\mathcal{I}_h^\sigma\right) \\ \mathcal{I}_h^\sigma = \left\{ \mathrm{median}\left(I_h^{\sigma, M_{nuc}}(z = Z)\right) \right\}_{Z \in [0, Z_{max}]} \end{cases}, \tag{17}$$

with MAD the median absolute deviation and $Z_{max}$ the depth of the last z-slice. *Figure 5—figure supplement 1f* shows the decay profile of a sample normalized using different values of $\sigma$, as well as the corresponding Gaussian-averaged nuclei fields.

With the aim of comparing normalized signals across different samples acquired with the same imaging setup, we further rescale the normalized signal by a global multiplicative factor designed to mitigate the sample-dependent impact of the normalizing factor $S_{h,0}^{r(\lambda_{blue}, \lambda_b)}$ in *Equation 16*. We multiply each normalized signal by a multiplicative factor such that the median intensity of the raw Hoechst signal computed in a region of brightest intensity is transmitted to the normalized Hoechst signal. The rationale is that the median intensity in the brightest regions reflects the expected intensity in the absence of artifacts, e.g., in outer cell layers close to the cover glass.

We tested the normalization method on a sample stained with both Hoechst and a Far-red nuclei stain, using Hoechst to normalize the Far-red channel and then look at the intensity profile. *Figure 5—figure supplement 1e* shows the depth decay for the unnormalized signal, as well as the signal normalized using the ratio $r(\lambda_{blue}, \lambda_{far-red}) = 0.48$ as found experimentally. On the same plot, we show the profile for the ratio $r = 1$ that would model a situation where we do not correct for wavelength dependency. In the latter, the intensity is over-corrected in depth, whereas with the correct ratio, we recover a flat profile in depth. Because there is an error associated with the measurement of $r$, we tested small variations around the value 0.48. The depth decay changes slightly, as shown on the right panel with a 10% difference above and below. XZ profiles for these two panels are shown on the right. In addition, we correct the Hoechst for its own intensity decay, in that case using $r = 1$ which recovers a flat profile in depth, this is shown *Figure 5—figure supplement 1g* with the corresponding XZ views. The last sanity check is performed by plotting the profile not only along the Z axis but also along X and Y, to ensure the absence of any other gradients of intensity. This is *Figure 5—figure supplement 1h*, plotted on the Far-red channel normalized with $r = 0.48$.

## Density maps

To quantify the local abundance of a target object, e.g., nuclei or specifically mitotic nuclei, we study three spatial fields: the density (number of objects per volume), the volume fraction (percentage of local space filled by the objects), and the map of objects volumes.

All fields are first computed at the voxel scale. The voxel-scale density field and map of objects volumes are computed by replacing each object by a single voxel of value, respectively, 1 or the object's volume placed at its centroid. The voxel-scale volume fraction corresponds to a binary mask of the target objects, and can be obtained directly with semantic segmentation methods or by binarising segmented object instances as output by Stardist3D. The voxel-scale fields are finally convolved with a Gaussian kernel of the desired analysis scale.

These quantities give a complete description of the local object abundance, as the volume fraction and map of objects volumes inform on the impact of volume on the object density. For instance, a large spatial region can misleadingly display a constant volume fraction but have large variations in object density and local objects volumes if the latter two compensate.

## Morphometric analysis

To quantify object-scale deformation, we fit an ellipse to each segmented object. This can be done efficiently by computing the inertia tensor $I$. For a given object composed of $N$ voxels at positions $\{\vec{x}_1, \vec{x}_2, \ldots, \vec{x}_N\}$, we first compute the positions of the voxels relative to the object's centroid $\vec{c}$, $\{\vec{r}_1, \vec{r}_2, \ldots, \vec{r}_N\}$, with $\vec{r}_k = \vec{x}_k - \vec{c} = (r_{k,1}, r_{k,2}, r_{k,3})$. The inertia tensor is then defined as:

$$I_{i,j} = \sum_{k=1}^{N} (\delta_{i,j} r_k^2 - r_{k,i} r_{k,j}), \tag{18}$$

with $r_k^2 = r_{k,1}^2 + r_{k,2}^2 + r_{k,3}^2$, and $\delta_{i,j}$ the Kronecker delta. The eigenvectors and eigenvalues of the inertia tensor correspond to the principal axes and moments of inertia of the object. Notably, the principal axes of $I$ are the same as the principal axes of the ellipse that best fits the object. The lengths of the ellipse's semi-axes can be obtained from the moments of inertia: if $(I_1, I_2, I_3)$ are the moments of inertia associated with the principal axes $(\vec{e}_1, \vec{e}_2, \vec{e}_3)$, the lengths of the corresponding semi-axii $(L_1, L_2, L_3)$ are given by:

$$L_i^2 = \frac{5}{2N}(I_1 + I_2 + I_3 - 2I_i), \quad i = 1, 2, 3. \tag{19}$$

Computing the inertia tensor for each object leads to a sparse tensor field that can be studied at arbitrary scales using the sparse Gaussian convolution equation defined previously. For instance, it can be used to study local object alignments by focusing only on the largest semi-axis (the main axis of deformation of the object). In practice, quantifying local object alignment by averaging inertia tensors introduces a bias by over-representing large objects. To avoid this bias, we instead compute the true strain tensor from the inertia tensor. The true strain tensor $T$ is defined as:

$$T = V \, diag(T_1, T_2, T_3) \, V^T, \tag{20}$$

with $V$ the matrix of eigenvectors of $I$ and $T_i = \frac{1}{3} \log \left( L_i^3 / (L_1 L_2 L_3) \right)$. The eigenvalues $(T_1, T_2, T_3)$ of $T$ represent the relative amount of deviation of each principal length from the radius of a sphere with the same volume as the object. Due to its relative nature, the true strain tensor is suited to compare the deformations of two objects with different sizes, and is thus appropriate for averaging in a local neighborhood.

## Acknowledgements

This work is supported by the French National Research Agency ('France 2030,' ANR-16-CONV-0001 from Excel lence Initiative of Aix-Marseille University- A*MIDEX), a generic grant to P-FL (ANR-19-CE13-0022), a generic grant to ST. ANR-22-CE30-0021 and an ERC grant (to P-FL, ERC SyG 101072123). We also acknowledge the France-Bioimaging Infrastructure (ANR-10-INBS-04). This work was supported by the Fondation pour la Recherche M´edicale, (to AG, grant number FDT202404018538, and to P-FL, grant number EQU202003010407) VDE acknowledges support by an HFSP long-term postdoctoral fellowship (HFSP LT0058/2022- L). ST and P-FL thank financial support from Inserm to the Booster Program Mecacell3D. We thank Dalia El-Arawi for providing one of the datasets used in *Figure 3—figure supplement 1*.

## Additional information

### Funding

| Funder | Grant reference number | Author |
| --- | --- | --- |
| Agence Nationale de la Recherche | ANR-16-CONV-0001 | Alice Gros<br>Jules Vanaret<br>Philippe Roudot<br>Pierre-François Lenne<br>Léo Guignard<br>Sham Tlili |
| Agence Nationale de la Recherche | ANR-19-CE13-0022 | Pierre-François Lenne |
| Agence Nationale de la Recherche | ANR-22-CE30-0021 | Sham Tlili |

| Funder | Grant reference number | Author |
|---|---|---|
| European Research Council | 10.3030/101072123 | Agathe Rostan<br>Pierre-François Lenne<br>Sham Tlili |
| Agence Nationale de la Recherche | ANR-10-INBS-04 | Pierre-François Lenne<br>Sham Tlili |
| Fondation pour la Recherche Médicale | FDT202404018538 | Alice Gros |
| Fondation pour la Recherche Médicale | EQU202003010407 | Pierre-François Lenne |
| Human Frontier Science Program | HFSP LT0058/2022- L | Valentin Dunsing-Eichenauer |
| Institut National de la Santé et de la Recherche Médicale | Inserm Booster Program MecaCell3D | Pierre-François Lenne<br>Sham Tlili |

The funders had no role in study design, data collection and interpretation, or the decision to submit the work for publication.

## Author contributions

Alice Gros, Resources, Data curation, Software, Formal analysis, Validation, Investigation, Visualization, Methodology, Writing – original draft, Writing – review and editing; Jules Vanaret, Resources, Data curation, Software, Formal analysis, Validation, Visualization, Methodology, Writing – original draft, Writing – review and editing; Valentin Dunsing-Eichenauer, Conceptualization, Resources, Data curation, Software, Formal analysis, Validation, Investigation, Visualization, Methodology, Writing – original draft; Agathe Rostan, Investigation; Philippe Roudot, Supervision, Funding acquisition; Pierre-François Lenne, Conceptualization, Supervision, Funding acquisition, Project administration; Léo Guignard, Conceptualization, Resources, Software, Formal analysis, Supervision, Funding acquisition, Validation, Methodology, Writing – original draft, Project administration, Writing – review and editing; Sham Tlili, Conceptualization, Resources, Formal analysis, Supervision, Funding acquisition, Validation, Investigation, Methodology, Writing – original draft, Project administration, Writing – review and editing

## Author ORCIDs

Alice Gros ⬤ https://orcid.org/0009-0000-8957-4673
Philippe Roudot ⬤ https://orcid.org/0000-0001-6632-8728
Pierre-François Lenne ⬤ https://orcid.org/0000-0003-1066-7506
Léo Guignard ⬤ https://orcid.org/0000-0002-3686-1385
Sham Tlili ⬤ https://orcid.org/0000-0001-6018-9923

Reviewer #1 (Public review): https://doi.org/10.7554/eLife.107154.3.sa1
Reviewer #2 (Public review): https://doi.org/10.7554/eLife.107154.3.sa2
Reviewer #3 (Public review): https://doi.org/10.7554/eLife.107154.3.sa3
Author response https://doi.org/10.7554/eLife.107154.3.sa4

# Additional files

## Supplementary files

MDAR checklist

## Data availability

All datasets used to reproduce the figures of the articles, the trained Stardist3D weights, and the associated training datasets can be found in the following link: https://doi.org/10.5281/zenodo.14748083. The Tapenade package is available on GitHub: https://github.com/GuignardLab/tapenade, *Vanaret et al., 2026*. The napari plugin napari-manual-registration is available here: https://github.com/GuignardLab/napari-manual-registration, *Vanaret and Gros, 2026a*. The napari plugin

napari-tapenade-processing is available here: https://github.com/GuignardLab/napari-tapenade-processing, *Vanaret and Gros, 2026b*. The napari plugin napari-spatial-correlation-plotter is available here: https://github.com/GuignardLab/napari-spatial-correlation-plotter, *Vanaret, 2026a*.

The following dataset was generated:

| Author(s) | Year | Dataset title | Dataset URL | Database and Identifier |
| --- | --- | --- | --- | --- |
| Vanaret J, Gros A, Dunsing V, Rostan A, Guignard L, Tlili SL | 2025 | Analysis Data and Stardist3D Trained Weights for Whole-mount Gastruloid Deep Imaging and Multiscale Analysis | https://doi.org/10.5281/zenodo.14748083 | Zenodo, 10.5281/zenodo.14748083 |

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
