## [Editor Report · eLife Assessment]

This work describes the establishment of an image analysis pipeline for signal correction, segmentation and quantitative data analysis of multilayered organoid and tumoroid systems. The revised study is **important** for the field to address many practical challenges in deep-tissue visualization. The image analysis pipeline is well-designed and **compelling**.

---

## [Referee Report · Reviewer #1 (Public review)]

Summary:

The image analysis pipeline is tested in analysing microscopy imaging data of gastruloids of varying sizes, for which an optimised protocol for in toto image acquisition is established based on whole mount sample preparation using an optimal refractive index matched mounting media, opposing dual side imaging with two-photon microscopy for enhaced laser penetration, dual view registration and weighted fusion for improved in toto sample data representation. For enhanced imaging speed in a two-photon microscope, parallel imaging was used and the authors performed spectral unmixing analysis to avoid issues of signal cross-talk.

In the image analysis pipeline image, different pre-treatments are done dependent on the analysis to be performed (for nuclear segmentation - contrast enhancement and normalisation; for quantitative analysis of gene expression - corrections for optical artifacts inducing signal intensity variations). Stardist3D was used for the nuclear segmentation. The study analyses in toto properties of gastruloid nuclear density, patterns of cell division, morphology, deformation and gene expression.

Strengths:

The methods developed are sound, well described and well validated, using a sample challenging for microscopy, gastruloids. Many of the established methods are very useful (e.g. registration, corrections, signal normalisation, lazy loading bioimage visualisation, spectral decomposition analysis), facilitate the development of quantitative research and would be of interest to the wide scientific community.

Comments on revisions:

I am happy with the job the authors have done with the revision. No further comments.

---

## [Referee Report · Reviewer #2 (Public review)]

Summary:

This study presents an integrated experimental and computational pipeline for high-resolution, quantitative imaging and analysis of gastruloids. The experimental module employs dual-view two-photon spectral imaging combined with optimized clearing and mounting techniques, enabling improved deep-tissue visualization compared with conventional methods. This advanced approach allows comprehensive 3D imaging of whole-mount immunostained gastruloids, capturing both tissue-scale architecture and single-cell-level information.

The computational module encompasses both pre-processing of acquired images and downstream analysis, providing quantitative insights into the structural and molecular characteristics of gastruloids. The pre-processing pipeline, tailored for dual-view two-photon microscopy, includes spectral unmixing of fluorescence signals using depth-dependent spectral profiles, as well as image fusion via rigid 3D transformation based on content-based block-matching algorithms. Nuclei segmentation was performed using a custom-trained StarDist3D model, validated against 2D manual annotations, and achieving an F1 score of 85+/-3% at a 50% intersection-over-union (IoU) threshold. Another custom-trained StarDist3D model enabled accurate detection of proliferating cells and the generation of 3D spatial maps of nuclear density and proliferation probability. Moreover, the pipeline facilitates detailed morphometric analysis of cell density and nuclear deformation, revealing pronounced spatial heterogeneities during early gastruloid morphogenesis.

All computational tools developed in this study are released as open-source, Python-based software.

Strengths:

The authors applied two-photon microscopy to whole-mount deep imaging of gastruloids, achieving in toto visualization at single-cell resolution. By combining spectral imaging with an unmixing algorithm, they successfully separated four fluorescent signals, enabling spatial analysis of gene expression patterns.

The image analysis method for nuclei segmentation was thoroughly benchmarked against existing methods, demonstrating advantages over conventional approaches, and its applicability across diverse datasets was convincingly established. The authors also evaluated the state-of-the-art Cellpose-SAM framework, showing that it performs well on their data and that the authors' preprocessing strategy can further enhance Cellpose-SAM's segmentation performance in deep tissues.

The entire computational workflow, from image pre-processing to segmentation with a custom-trained StarDist3D model and subsequent quantitative analysis, is made available as open-source software. In addition, user-friendly interfaces are provided through the open-source, community-driven napari platform, facilitating interactive exploration and analysis.

Weaknesses:

In my initial review, I noted that the developed image analysis pipeline lacked benchmarking against existing methods and provided only a limited demonstration of its applicability to other datasets. These points have been appropriately addressed in the revised manuscript, and I have no further weaknesses to note.

Appraisal:

The authors set out to establish a quantitative imaging and analysis pipeline for gastruloids using dual-view two-photon microscopy, spectral unmixing, and a custom computational framework for 3D segmentation and gene expression analysis. This aim was compellingly achieved. The integration of experimental and computational modules enables high-resolution in toto imaging and robust quantitative analysis at the single-cell level. The data presented support the authors' conclusions regarding the ability to capture spatial patterns of gene expression and cellular morphology across developmental stages.

Impact and utility:

This work presents a compelling and broadly applicable methodological advance. The approach is particularly impactful for the developmental biology community, as it allows researchers to extract quantitative information from high-resolution images to better understand morphogenetic processes. The data are publicly available on Zenodo, and the software is released on GitHub, making them highly valuable resources for the community. Given that suitable datasets for developing advanced 3D cell segmentation methods remain scarce in biological image analysis, the public release of these data is significant and is expected to stimulate further advances in the development of sophisticated computational approaches.

Comments on revisions:

The authors have addressed the previous revision thoroughly and appropriately. I have no further suggestions or additional recommendations at this time.

---

## [Referee Report · Reviewer #3 (Public review)]

Summary

The paper presents a imaging and analysis pipeline for whole-mount gastruloid imaging with two-photon microscopy. The presented pipeline includes spectral unmixing, registration, segmentation, and a wavelength-depended intensity normalization step, followed by quantitative analysis of spatial gene expression patterns and nuclear morphometry on a tissue level. The utility of the approach is demonstrated by several experimental findings such as establishing spatial correlations between local nuclear deformation and tissue density changes, as well as radial distribution pattern of mesoderm markers. The pipeline is distributed as a Python package, notebooks and multiple napari plugins.

Strengths

The paper is well-written with detailed methodological descriptions, which I think would make it a valuable reference for researchers performing similar volumetric tissue imaging experiments (gastruloids/organoids). The pipeline itself addresses many practical challenges including resolution loss within tissue, registration of large volumes, nuclear segmentation, and intensity normalization. Especially the intensity decay measurements and wavelength-dependent intensity normalization approach using nuclear (Hoechst) signal as reference is very interesting and should be applicable to other imaging contexts. The morphometric analysis is equally well done with the correlation between nuclear shape deformation and tissue density changes being a interesting finding. The paper is quite thorough in its technical description of the methods (which are a lot) and their experimental validation is appropriate. Finally, the provided code and napari plugins seem to be well done (I installed a selected list of the plugins and they ran without issues) and should be very helpful for the community.

Comments on revisions:

The minor issues that I originally raised in my first review have been fully resolved in the revised version.

---

## [Author Response]

The following is the authors’ response to the original reviews.

**Public Reviews:**

**Reviewer #1 (Public review):**
Summary:The image analysis pipeline is tested in analysing microscopy imaging data of gastruloids of varying sizes, for which an optimised protocol for in toto image acquisition is established based on whole mount sample preparation using an optimal refractive index matched mounting media, opposing dual side imaging with two-photon microscopy for enhanced laser penetration, dual view registration, and weighted fusion for improved in toto sample data representation. For enhanced imaging speed in a two-photon microscope, parallel imaging was used, and the authors performed spectral unmixing analysis to avoid issues of signal cross-talk.In the image analysis pipeline, different pre-treatments are done depending on the analysis to be performed (for nuclear segmentation - contrast enhancement and normalisation; for quantitative analysis of gene expression - corrections for optical artifacts inducing signal intensity variations). Stardist3D was used for the nuclear segmentation. The study analyses into properties of gastruloid nuclear density, patterns of cell division, morphology, deformation, and gene expression.Strengths:The methods developed are sound, well described, and well-validated, using a sample challenging for microscopy, gastruloids. Many of the established methods are very useful (e.g. registration, corrections, signal normalisation, lazy loading bioimage visualisation, spectral decomposition analysis), facilitate the development of quantitative research, and would be of interest to the wider scientific community.

We thank the reviewer for this positive feedback.

Weaknesses:A recommendation should be added on when or under which conditions to use this pipeline.

We thank the reviewer for this valuable feedback, we added the text in the revised version, ines 418 to 474. “In general, the pipeline is applicable to any tissue, but it is particularly useful for large and dense 3D samples—such as organoids, embryos, explants, spheroids, or tumors—that are typically composed of multiple cell layers and have a thickness greater than 50 µm”.

“The processing and analysis pipeline are compatible with any type of 3D imaging data (e.g. confocal, 2 photon, light-sheet, live or fixed)”.

“Spectral unmixing to remove signal cross-talk of multiple fluorescent targets is typically more relevant in two-photon imaging due to the broader excitation spectra of fluorophores compared to single-photon imaging. In confocal or light-sheet microscopy, alternating excitation wavelengths often circumvents the need for unmixing. Spectral decomposition performs even better with true spectral detectors; however, these are usually not non-descanned detectors, which are more appropriate for deep tissue imaging. Our approach demonstrates that simultaneous cross-talk-free four-color two-photon imaging can be achieved in dense 3D specimen with four non-descanned detectors and co-excitation by just two laser lines. Depending on the dispersion in optically dense samples, depth-dependent apparent emission spectra need to be considered”.

“Nuclei segmentation using our trained StarDist3D model is applicable to any system under two conditions: (1) the nuclei exhibit a star-convex shape, as required by the StarDist architecture, and (2) the image resolution is sufficient in XYZ to allow resampling. The exact sampling required is object- and system-dependent, but the goal is to achieve nearly isotropic objects with diameters of approximately 15 pixels while maintaining image quality. In practice, images containing objects that are natively close to or larger than 15 pixels in diameter should segment well after resampling. Conversely, images with objects that are significantly smaller along one or more dimensions will require careful inspection of the segmentation results”.

“Normalization is broadly applicable to multicolor data when at least one channel is expected to be ubiquitously expressed within its domain. Wavelength-dependent correction requires experimental calibration using either an ubiquitous signal at each wavelength. Importantly, this calibration only needs to be performed once for a given set of experimental conditions (e.g., fluorophores, tissue type, mounting medium)”.

“Multi-scale analysis of gene expression and morphometrics is applicable to any 3D multicolor image. This includes both the 3D visualization tools (Napari plugins) and the various analytical plots (e.g., correlation plots, radial analysis). Multi-scale analysis can be performed even with imperfect segmentation, as long as segmentation errors tend to cancel out when averaged locally at the relevant spatial scale. However, systematic errors—such as segmentation uncertainty along the Z-axis due to strong anisotropy—may accumulate and introduce bias in downstream analyses. Caution is advised when analyzing hollow structures (e.g., curved epithelial monolayers with large cavities), as the pipeline was developed primarily for 3D bulk tissues, and appropriate masking of cavities would be needed”.

**Reviewer #2 (Public review):**
Summary:This study presents an integrated experimental and computational pipeline for high-resolution, quantitative imaging and analysis of gastruloids. The experimental module employs dual-view two-photon spectral imaging combined with optimized clearing and mounting techniques to image whole-mount immunostained gastruloids. This approach enables the acquisition of comprehensive 3D images that capture both tissue-scale and single-cell level information.The computational module encompasses both pre-processing of acquired images and downstream analysis, providing quantitative insights into the structural and molecular characteristics of gastruloids. The pre-processing pipeline, tailored for dual-view two-photon microscopy, includes spectral unmixing of fluorescence signals using depth-dependent spectral profiles, as well as image fusion via rigid 3D transformation based on content-based block-matching algorithms. Nuclei segmentation was performed using a custom-trained StarDist3D model, validated against 2D manual annotations, and achieving an F1 score of 85+/-3% at a 50% intersection-over-union (IoU) threshold. Another custom-trained StarDist3D model enabled accurate detection of proliferating cells and the generation of 3D spatial maps of nuclear density and proliferation probability. Moreover, the pipeline facilitates detailed morphometric analysis of cell density and nuclear deformation, revealing pronounced spatial heterogeneities during early gastruloid morphogenesis.All computational tools developed in this study are released as open-source, Python-based software.Strengths:The authors applied two-photon microscopy to whole-mount deep imaging of gastruloids, achieving in toto visualization at single-cell resolution. By combining spectral imaging with an unmixing algorithm, they successfully separated four fluorescent signals, enabling spatial analysis of gene expression patterns.The entire computational workflow, from image pre-processing to segmentation with a custom-trained StarDist3D model and subsequent quantitative analysis, is made available as open-source software. In addition, user-friendly interfaces are provided through the open-source, community-driven Napari platform, facilitating interactive exploration and analysis.

We thank the reviewer for this positive feedback.

Weaknesses:The computational module appears promising. However, the analysis pipeline has not been validated on datasets beyond those generated by the authors, making it difficult to assess its general applicability.

We agree that applying our analysis pipeline to published datasets—particularly those acquired with different imaging systems—would be valuable. However, only a few high-resolution datasets of large organoid samples are publicly available, and most of these either lack multiple fluorescence channels or represent 3D hollow structures. Our computational pipeline consists of several independent modules: spectral filtering, dual-view registration, local contrast enhancement, 3D nuclei segmentation, image normalization based on a ubiquitous marker, and multiscale analysis of gene expression and morphometrics. We added the following sentences to the Discussion, lines 418 to 474, and completed the discussion on applicability with a table showing the purpose, requirements, applicability and limitations of each step of the processing and analysis pipeline.

“Spectral filtering has already been applied in other systems (e.g. [7] and [8]), but is here extended to account for imaging depth-dependent apparent emission spectra of the different fluorophores. In our pipeline, we provide code to run spectral filtering on multichannel images, integrated in Python. In order to apply the spectral filtering algorithm utilized here, spectral patterns of each fluorophore need to be calibrated as a function of imaging depth, which depend on the specific emission windows and detector settings of the microscope”.

“Image normalization using a wavelength-dependent correction also requires calibration on a given imaging setup to measure the difference in signal decay among the different fluorophores species. To our knowledge, the calibration procedures for spectral-filtering and our image-normalization approach have not been performed previously in 3D samples, which is why validation on published datasets is not readily possible. Nevertheless, they are described in detail in the Methods section, and the code used—from the calibration measurements to the corrected images—is available open-source at the Zenodo link in the manuscript”.

Dual-view registration, local contrast enhancement, and multiscale analysis of gene expression and morphometrics are not limited to organoid data or our specific imaging modalities. To evaluate our 3D nuclei segmentation model, we tested it on diverse systems, including gastruloids stained with the nuclear marker Draq5 from Moos et al. [1]; breast cancer spheroids; primary ductal adenocarcinoma organoids; human colon organoids and HCT116 monolayers from Ong et al. [2]; and zebrafish tissues imaged by confocal microscopy from Li et al [3]. These datasets were acquired using either light-sheet or confocal microscopy, with varying imaging parameters (e.g., objective lens, pixel size, staining method). The results are added in the manuscript, Fig. S9b.

Besides, the nuclei segmentation component lacks benchmarking against existing methods.

We agree with the reviewer that a benchmark against existing segmentation methods would be very useful. We tried different pre-trained models:

CellPose, which we tested in a previous paper ([4]) and which showed poor performances compared to our trained StarDist3D model.

DeepStar3D ([2]) is only available in the software 3DCellScope. We could not benchmark the model on our data, because the free and accessible version of the software is limited to small datasets. An image of a single whole-mount gastruloid with one channel, having dimensions (347,467,477) was too large to be processed, see screenshot below. The segmentation model could not be extracted from the source code and tested externally because the trained DeepStar3D weights are encrypted.

**Author response image 1. sa4fig1:** Screenshot of the 3DCellScore software. We could not perform 3D nuclei segmentation of a whole-mount gastruloids because the image size was too large to be processed.

AnyStar ([5]), which is a model trained from the StarDist3D architecture, was not performing well on our data because of the heterogeneous stainings. Basic pre-processing such as median and gaussian filtering did not improve the results and led to wrong segmentation of touching nuclei. AnyStar was demonstrated to segment well colon organoids in Ong et al, 2025 ([2]), but the nuclei were more homogeneously stained. Our Hoechst staining displays bright chromatin spots that are incorrectly labeled as individual nuclei.

Cellos ([6]), another model trained from StarDist3D, was also not performing well. The objects used for training and to validate the results are sparse and not touching, so the predicted segmentation has a lot of false negatives even when lowering the probability threshold to detect more objects. Additionally, the network was trained with an anisotropy of (9,1,1), based on images with low z resolution, so it performed poorly on almost isotropic images. Adapting our images to the network’s anisotropy results in an imprecise segmentation that can not be used to measure 3D nuclei deformations.

We tried both Cellos and AnyStar predictions on a gastruloid image from Fig. S2 of our main manuscript. The results are added in the manuscript, Fig. S9b. Fig3 displays the results qualitatively compared to our trained model Stardist-tapenade.

**Author response image 2. sa4fig2:** Qualitative comparison of two published segmentation models versus our model. We show one slice from the XY plane for simplicity. Segmentations are displayed with their contours only. (Top left) Gastruloid stained with Hoechst, image extracted from Fig S2 of our manuscript. (Top right) Same image overlayed with the prediction from the Cellos model, showing many false negatives. (Bottom left) Same image overlayed with the prediction from our Stardist-tapenade model. (Bottom right) Same image overlayed with the prediction from the AnyStar model, false positives are indicated with a red arrow.

CellPose-SAM, which is a recent model developed building on the CellPose framework. The pre-trained model performs well on gastruloids imaged using our pipeline, and performs better than StarDist3D at segmenting elongated objects such as deformed nuclei. The performances are qualitatively compared on Fig. S9a and S10. We also demonstrate how using local contrast enhancement improves the results of CellPose-SAM (Fig. S10a), showing the versatility of the Tapenade pre-processing module. Tissue-scale, packing-related metrics from Cellpose–SAM labels qualitatively match those from stardist-tapenade as shown Fig.10c and d.

Appraisal:The authors set out to establish a quantitative imaging and analysis pipeline for gastruloids using dual-view two-photon microscopy, spectral unmixing, and a custom computational framework for 3D segmentation and gene expression analysis. This aim is largely achieved. The integration of experimental and computational modules enables high-resolution in toto imaging and robust quantitative analysis at the single-cell level. The data presented support the authors' conclusions regarding the ability to capture spatial patterns of gene expression and cellular morphology across developmental stages.Impact and utility:This work presents a compelling and broadly applicable methodological advance. The approach is particularly impactful for the developmental biology community, as it allows researchers to extract quantitative information from high-resolution images to better understand morphogenetic processes. The data are publicly available on Zenodo, and the software is released on GitHub, making them highly valuable resources for the community.

We thank the reviewer for these positive feedbacks.

**Reviewer #3 (Public review):**
SummaryThe paper presents an imaging and analysis pipeline for whole-mount gastruloid imaging with two-photon microscopy. The presented pipeline includes spectral unmixing, registration, segmentation, and a wavelength-dependent intensity normalization step, followed by quantitative analysis of spatial gene expression patterns and nuclear morphometry on a tissue level. The utility of the approach is demonstrated by several experimental findings, such as establishing spatial correlations between local nuclear deformation and tissue density changes, as well as the radial distribution pattern of mesoderm markers. The pipeline is distributed as a Python package, notebooks, and multiple napari plugins.StrengthsThe paper is well-written with detailed methodological descriptions, which I think would make it a valuable reference for researchers performing similar volumetric tissue imaging experiments (gastruloids/organoids). The pipeline itself addresses many practical challenges, including resolution loss within tissue, registration of large volumes, nuclear segmentation, and intensity normalization. Especially the intensity decay measurements and wavelength-dependent intensity normalization approach using nuclear (Hoechst) signal as reference are very interesting and should be applicable to other imaging contexts. The morphometric analysis is equally well done, with the correlation between nuclear shape deformation and tissue density changes being an interesting finding. The paper is quite thorough in its technical description of the methods (which are a lot), and their experimental validation is appropriate. Finally, the provided code and napari plugins seem to be well done (I installed a selected list of the plugins and they ran without issues) and should be very helpful for the community.

We thank the reviewer for his positive feedback and appreciation of our work.

WeaknessesI don't see any major weaknesses, and I would only have two issues that I think should be addressed in a revision:(1) The demonstration notebooks lack accompanying sample datasets, preventing users from running them immediately and limiting the pipeline's accessibility. I would suggest to include (selective) demo data set that can be used to run the notebooks (e.g. for spectral unmixing) and or provide easily accessible demo input sample data for the napari plugins (I saw that there is some sample data for the processing plugin, so this maybe could already be used for the notebooks?).

We thank the reviewer for this relevant suggestion. The 7 notebooks were updated to automatically download sample tests. The different parts of the pipeline can now be run immediately:

https://github.com/GuignardLab/tapenade/tree/chekcs_on_notebooks/src/tapenade/notebooks

(2) The results for the morphometric analysis (Figure 4) seem to be only shown in lateral (xy) views without the corresponding axial (z) views. I would suggest adding this to the figure and showing the density/strain/angle distributions for those axial views as well.

A morphometric analysis based on the axial views was added as Fig. S6a of the manuscript, complementary to the XY views.

**Recommendations for the authors:**

**Reviewer #1 (Recommendations for the authors):**
In lines 64 and 65, it is mentioned that confocal and light-sheet microscopy remain limited to samples under 100μm in diameter. I would recommend revising this sentence. In the paper of Moos and colleagues (also cited in this manuscript; PMID: 38509326), gastruloid samples larger than 100μm are imaged in toto with an open-top dual-view and dual-illumination light-sheet microscope, and live cell behaviour is analysed. Another example, if considering also multi-angle systems, is the impressive work of McDole and colleagues (PMID: 30318151), in which one of the authors of this manuscript is a corresponding author. There, multi-angle light sheet microscopy is used for in toto imaging and reconstruction of post-implantation mouse development (samples much larger than 100μm). Some multi-sample imaging strategies have been developed for this type of imaging system, though not to the sample number extent allowed by the Viventis LS2 system or the Bruker TruLive3D imager, which have higher image quality limitations.

We thank the reviewer for this remark. As reported in their paper, Moos *et al.* used dual-view light-sheet microscopy to image gastruloids, which are particularly dense and challenging tissues, with whole-mount samples of approximately 250 µm in diameter. Nevertheless, their image quality metric (DCT) shows a rapid twofold decrease within 50 µm depth (Extended Fig 5.h), whereas with two-photon microscopy, our image quality metric (FRC-QE) decreases by a factor of two over 150 µm in non-cleared samples (PBS) (see Fig. 2 c). While these two measurements (FRC-QE versus DCT) are not directly comparable, the observed difference reflects the superior depth performance of two-photon microscopy, owing in part to the use of non-descanned detectors. In our case, imaging was performed with Hoechst, a blue fluorophore suboptimal for deep imaging, whereas in the Moos dataset (Draq5, far-red), the configuration was more favorable for imaging in depth which further supports our conclusion.

In McDole et al, tissues reaching 250µm were imaged from 4 views, but do not reach cellular-scale resolution in deeper layers compatible with cell segmentation to our knowledge.

We corrected the sentence ‘However, light-sheet and confocal imaging approaches remain limited to relatively small organoids typically under 100 micrometers in diameter ‘ by the following (line 64) :

“While advances in light-sheet microscopy have extended imaging depth in organoids, maintaining high image quality throughout thick samples remains challenging. In practice, quantitative analyses are still largely restricted to organoids under roughly 100 µm in diameter”.

It is worth mentioning that two-photon microscopes are much more widely available than light sheet microscopes, and light sheet systems with 2-photon excitation are even less accessible, which makes the described workflow of Gros and colleagues have a wide community interest.

We thank the reviewer for this remark, and added this suggestion line 74:

“Finally, two-photon microscopes are typically more accessible than light-sheet systems and allow for straightforward sample mounting, as they rely on procedures comparable to standard confocal imaging”.

**Reviewer #2 (Recommendations for the authors):**
Suggestions:A comparison with established pre-trained models for 3D organoid image segmentation (e.g., Cellos[1], AnyStar[2], and DeepStar3D[3], all based on StarDist3D) would help highlight the advantages of the authors' custom StarDist3D model, which has been specifically optimized for two-photon microscopy images.(1) Cellos: https://doi.org/10.1038/s41467-023-44162-6(2) AnyStar: https://doi.org/10.1109/WACV57701.2024.00742(3) DeepStar3D: https://doi.org/10.1038/s41592-025-02685-4

We agree with the reviewer that a benchmark against existing segmentation methods is very useful. This is addressed in the revised version, as detailed above (Figure 3).

Recommendations:Please clarify the following point. In line 195, the authors state, "This allowed us to detect all mitotic nuclei in whole-mount samples for any stage and size." Does this mean that the custom-trained StarDist3D model can detect 100% of mitotic nuclei? It was not clear from the manuscript, figures, or videos how this was validated. Given the reported performance scores of the StarDist3D model for detecting all nuclei, claiming 100% detection of mitotic nuclei seems surprisingly high.

We thank the reviewer for this comment. As it was detailed in the methods section, the detection score reaches 82%, and only the complete pipeline (detection+minimal manual curation) allows us to detect all mitotic nuclei. To make it clearer, the following precisions were added in the Results section:

”To detect division events, we stained gastruloids with phosphohistone H3 (ph3) and trained a separate custom Stardist3D model using 3D annotations of nuclei expressing ph3 (see Methods III H). This model together allowed us to detect nearly all mitotic nuclei in whole-mount samples for any stage and size (Fig.3f and Suppl.Movie 4), and we used minimal manual curation to correct remaining errors.”

Minor corrections:It appears that Figures 4-6 are missing from the submitted version, but they can be found in the manuscript available on bioRxiv.

We thank the reviewer for this remark, this was corrected immediately to add Figures 4 to 6.

In line 185, is the intended phrase "by comparing the 2D predictions and the 2D sliced annotated segments..."?

To gain some clarity, we replaced the initial sentence:

“The f1 score obtained by comparing the 3D prediction and the 3D ground-truth is well approximated by the f1 score obtained by comparing the 2D annotations and the 2D sliced annotated segments, with at most a 5% difference between the two scores.” by

“The f1 score obtained in 3D (3D prediction compared with the 3D ground-truth) is well approximated by the f1 score obtained in 2D (2D predictions compared with the 2D sliced annotated segments). The difference between the 2 scores was at most 5%.”

**Reviewer #3 (Recommendations for the authors):**
(1) How is the "local neighborhood volume" defined, and how was it computed?

The reviewer is referring to this paragraph (the term is underscored) :

“To probe quantities related to the tissue structure at multiple scales, we smooth their signal with a Gaussian kernel of width σ, with σ defined as the spatial scale of interest. From the segmented nuclei instances, we compute 3D fields of cell density (number of cells per unit volume), nuclear volume fraction (ratio of nuclear volume to local neighborhood volume), and nuclear volume at multiple scales.”

To improve clarity, the phrasing has been revised: the term *local neighborhood volume* has been replaced by *local averaging volume*, and a reference to the Methods section has been added.

From the segmented nuclei instances, we compute 3D fields of cell density (number of cells per unit volume), nuclear volume fraction (ratio of space occupied by nuclear volume within the local averaging volume, as defined in the Methods III I), and nuclear volume at multiple scales.

(2) In the definition of inertia tensor (18), isn't the inner part normally defined in the reversed way (delta_i,j - ...)?

We thank the reviewer for noticing this error, which we fixed in the manuscript.

(3) For intensity normalization, the paper uses the Hoechst signal density as a proxy for a ubiquitous nuclei signal. I would assume that this is problematic, for eg, dividing cells (which would overestimate it). Would using the average Hoechst signal per nucleus mask (as segmentation is available) be a better proxy?

We agree that this idea is appealing if one assumes a clear relationship between nuclear volume and Hoechst intensity. However, since cell and nuclear volumes vary substantially with differentiation state (see Fig. 4), such a normalization approach would introduce additional biases at large spatial scales. We believe that the most robust improvement would instead consist in masking dividing cells during the normalization procedure, as these events could be detected and excluded from the computation.

Nonetheless, we believe the method proposed by the reviewer could prove relevant for other types of data, so we will implement this recommendation in the code available in the Tapenade package.

(4) Figures 4-6 were part of the Supplementary Material, but should be included in the main text?

We thank the reviewer for this remark, this was corrected immediately to add Figures 4-6.

We also noticed a missing reference to Fig. S3 in the main text, so we added lines 302 to 307 to comment on the wavelength-dependency of the normalization method. We improved the description of Fig.6, which lacked clarity (line 316 to 321, line 327).

(1) Moos, F., Suppinger, S., de Medeiros, G., Oost, K.C., Boni, A., Rémy, C., Weevers, S.L., Tsiairis, C., Strnad, P. and Liberali, P., 2024. Open-top multisample dual-view light-sheet microscope for live imaging of large multicellular systems. *Nature Methods*, *21*(5), pp.798-803.

(2) Ong, H. T.; Karatas, E.; Poquillon, T.; Grenci, G.; Furlan, A.; Dilasser, F.; Mohamad Raffi, S. B.; Blanc, D.; Drimaracci, E.; Mikec, D.; Galisot, G.; Johnson, B. A.; Liu, A. Z.; Thiel, C.; Ullrich, O.; OrgaRES Consortium; Racine, V.; Beghin, A. (2025). Digitalized organoids: integrated pipeline for high-speed 3D analysis of organoid structures using multilevel segmentation and cellular topology*. Nature Methods*, *22*(6), pp.1343-1354

(3) Li, L., Wu, L., Chen, A., Delp, E.J. and Umulis, D.M., 2023. 3D nuclei segmentation for multi-cellular quantification of zebrafish embryos using NISNet3D. *Electronic Imaging*, *35*, pp.1-9.

(4) Vanaret, J., Dupuis, V., Lenne, P. F., Richard, F., Tlili, S., & Roudot, P. (2023). A detector-independent quality score for cell segmentation without ground truth in 3D live fluorescence microscopy. *IEEE Journal of Selected Topics in Quantum Electronics*, *29*(4:Biophotonics), 1-12.

(5) Dey, N., Abulnaga, M., Billot, B., Turk, E. A., Grant, E., Dalca, A. V., & Golland, P. (2024). AnyStar: Domain randomized universal star-convex 3D instance segmentation. In *Proceedings of the IEEE/CVF Winter Conference on Applications of Computer Vision* (pp. 7593-7603).

(6) Mukashyaka, P., Kumar, P., Mellert, D. J., Nicholas, S., Noorbakhsh, J., Brugiolo, M., ... & Chuang, J. H. (2023). High-throughput deconvolution of 3D organoid dynamics at cellular resolution for cancer pharmacology with Cellos. *Nature Communications*, *14*(1), 8406.

(7) Rakhymzhan, A., Leben, R., Zimmermann, H., Günther, R., Mex, P., Reismann, D., ... & Niesner, R. A. (2017). Synergistic strategy for multicolor two-photon microscopy: application to the analysis of germinal center reactions in vivo. Scientific reports, 7(1), 7101.

(8) Dunsing, V., Petrich, A., & Chiantia, S. (2021). Multicolor fluorescence fluctuation spectroscopy in living cells via spectral detection. Elife, 10, e69687.